# The East Asian gut microbiome is distinct from colocalized White subjects and connected to metabolic health

Qi Yan Ang[1†], Diana L Alba[2,3†], Vaibhav Upadhyay[1†], Jordan E Bisanz[1], Jingwei Cai[4], Ho Lim Lee[2], Eliseo Barajas[2], Grace Wei[2], Cecilia Noecker[1], Andrew D Patterson[4], Suneil K Koliwad[2,3*], Peter J Turnbaugh[1*]

[1]Department of Microbiology and Immunology, G.W. Hooper Research Foundation, San Francisco, United States; [2]Diabetes Center, University of California San Francisco, San Francisco, United States; [3]Division of Endocrinology, Diabetes, and Metabolism, Department of Medicine, University of California San Francisco, San Francisco, United States; [4]Center for Molecular Toxicology and Carcinogenesis, Department of Veterinary & Biomedical Sciences, Pennsylvania State University, College Park, United States

**Abstract** East Asians (EAs) experience worse metabolic health outcomes compared to other ethnic groups at lower body mass indices; however, the potential role of the gut microbiota in contributing to these health disparities remains unknown. We conducted a multi-omic study of 46 lean and obese East Asian and White participants living in the San Francisco Bay Area, revealing marked differences between ethnic groups in bacterial richness and community structure. White individuals were enriched for the mucin-degrading *Akkermansia muciniphila*. East Asian subjects had increased levels of multiple bacterial phyla, fermentative pathways detected by metagenomics, and the short-chain fatty acid end-products acetate, propionate, and isobutyrate. Differences in the gut microbiota between the East Asian and White subjects could not be explained by dietary intake, were more pronounced in lean individuals, and were associated with current geographical location. Microbiome transplantations into germ-free mice demonstrated stable diet- and host genotype-independent differences between the gut microbiotas of East Asian and White individuals that differentially impact host body composition. Taken together, our findings add to the growing body of literature describing microbiome variations between ethnicities and provide a starting point for defining the mechanisms through which the microbiome may shape disparate health outcomes in East Asians.

**\*For correspondence:**
Suneil.Koliwad@ucsf.edu (SKK);
Peter.Turnbaugh@ucsf.edu (PJT)

†These authors contributed equally

## Introduction

Culture-independent surveys have emphasized differences in gut microbial community structure between countries (*Hehemann et al., 2010*; *Vangay et al., 2018*; *Yatsunenko et al., 2012*), but the factors that contribute to these differences are poorly understood. Diet is a common hypothesis for geographical variations in the gut microbiota (*De Filippo et al., 2010*; *Devoto et al., 2019*), based upon extensive data from intervention experiments in humans and mouse models (*Bisanz et al., 2019*; *Carmody et al., 2015*; *David et al., 2014*; *Gehrig et al., 2019*). However, diet is just one of the many factors that distinguishes human populations at the global scale, motivating the desire for a more holistic approach. Self-identified race/ethnicity (SIRE) provides a useful alternative, as it integrates the broader national or cultural tradition of a given social group and is closely tied to both dietary intake and genetic ancestry. Multiple studies have reported associations between the gut microbiota

**eLife digest** The community of microbes living in the human gut varies based on where a person lives, in part because of differences in diets but also due to factors still incompletely understood. In turn, this 'microbiome' may have wide-ranging effects on health and diseases such as obesity and diabetes.

Many scientists want to understand how differences in the microbiome emerge between people, and whether this may explain why certain diseases are more common in specific populations. Self-identified race or ethnicity can be a useful tool in that effort, as it can serve as a proxy for cultural habits (such as diets) or genetic information.

In the United States, self-identified East Asian Americans often have worse 'metabolic health' (e.g. levels of sugar or certain fat molecules in the blood) at a lower weight than those identifying as White. Ang, Alba, Upadhyay et al. investigated whether this health disparity was linked to variation in the gut microbiome. Samples were collected from 46 lean and obese individuals living in the San Francisco Bay Area who identified as White or East Asian.

The analyses showed that while the gut microbiome of White participants changed in association with obesity, the microbiomes of East Asian participants were distinct from their White counterparts even at normal weight, with features mirroring what was seen in White individuals in the context of obesity. Although these differences were connected to people's current address, they were not attributable to dietary differences.

Ang, Alba, Upadhyay et al. then transplanted the microbiome of the participants into genetically identical mice with microbe-free guts. The differences between the gut microbiomes of White and East Asian participants persisted in recipient animals. When fed the same diet, the mice also gained different amounts of weight depending on the ethnic identity of the microbial donor.

These results show that self-identified ethnicity may be an important variable to consider in microbiome studies, alongside other factors such as geography. Ultimately, this research may help to design better, more personalized treatments for an array of conditions.

and ethnicity in China (*Khine et al., 2019*), the Netherlands (*Deschasaux et al., 2018*), Singapore (*Xu et al., 2020*), and the United States (*Brooks et al., 2018*; *Sordillo et al., 2017*). In contrast, a recent study of Asian immigrants suggested that once an individual relocates to a new country, the microbiota rapidly assumes the structure of the country of residence (*Vangay et al., 2018*). Thus, the degree to which microbiome signatures of ethnicity persist following immigration and their consequences for host pathophysiology remain an open question.

The links between ethnicity and metabolic disease are well established. For example, East Asian (EA) subjects are more likely to develop health-related metabolic complications at lower body mass index (BMI) compared to their White (W) counterparts (*Gu et al., 2006*; *Zheng et al., 2011*). Moreover, Asian Americans have persistent ethnic differences in metabolic phenotypes following immigration (*Jih et al., 2014*), including a decoupling of BMI from total body fat percentage (*Alba et al., 2018*). The mechanisms contributing to these ethnic differences in fat accrual remain unknown. Human genetic polymorphisms may play a role (*Wen et al., 2010*; *Xiang et al., 2004*); however, putative alleles are often shared between members of different ethnic groups (*Gravel et al., 2011*). The gut microbiome might offer a possible explanation for differences in metabolic disease rates across ethnic groups (*He et al., 2018*), but there has been a relative scarcity of microbiome studies in this area (*Gaulke and Sharpton, 2018*).

These observations led us to hypothesize that ethnicity-associated differences in host metabolic phenotypes may be determined by corresponding differences in the gut microbiome. First, we sought to better understand the extent to which ethnicity is linked to the human gut microbiome in states of health and disease. We conducted a cross-sectional multi-omic analysis of the gut microbiome using paired 16S rRNA gene sequencing (16S-seq), metagenomics, and metabolomics from the Inflammation, Diabetes, Ethnicity, and Obesity (IDEO) cohort at the University of California, San Francisco. IDEO includes rich metabolic, dietary, and socioeconomic metadata (*Alba et al., 2018*), a restricted geographical distribution within the San Francisco Bay Area, and a balanced distribution of EA and W individuals that are both lean and obese (*Supplementary file 1A*). We report marked differences

in gut microbial richness, community structure, and metabolic end-products between EA and W individuals in the IDEO cohort. We then used microbiome transplantations to assess the stability of ethnicity-associated differences in the gut microbiota in the context of genetically identical mice fed the same diet. We also explored the functional consequences of these differences for host metabolic phenotypes. Our results emphasize the importance of considering ethnicity in microbiome research and further complicate prior links between metabolic disease and the gut microbiome (*Ley et al., 2006*; *Turnbaugh et al., 2008*; *Wu et al., 2020*), which may be markedly different across diverse ethnic groups.

## Results

Ethnicity was associated with inter-individual variations in the human gut microbiota. Principal coordinates analysis of PhILR Euclidean distances from 16S-seq data (*Supplementary file 1B*, n=22 EA, 24 W subjects) revealed a subtle but significant separation between the gut microbiotas of EA and W subjects (p=0.006, $R^2$=0.046, ADONIS; *Figure 1A*). Statistical significance was robust to the distance metric used (*Supplementary file 1C*). Bacterial diversity was significantly higher in W individuals across three distinct metrics: Faith's phylogenetic diversity, amplicon sequence variant (ASV) richness, and Shannon diversity (*Figure 1B*). Six bacterial phyla were significantly different between ethnicities (*Figure 1C*), of which only one phylum, *Verrucomicrobiota*, was significantly enriched in W subjects.

Phylogenetic analyses of all ASVs revealed marked variations in the direction of change across different phyla between EA and W subjects (*Figure 1—figure supplement 1A*), indicating that the phylum level trends (*Figure 1C*) resulted from the integration of subtle shifts across multiple component members (*Figure 1D–F*). Several significant differences were detectable at the genus level (*Figure 1D–E*), including *Blautia*, *Bacteroides,* and *Streptococcus* which were significantly enriched in EA subjects. We also identified two ASVs that were significantly different between ethnicities: *Blautia obeum* and a *Streptococcus* species, both enriched in EA subjects (*Figure 1F*). There were no significant differences between ethnicities in  16S rRNA copy number (*Figure 1—figure supplement 1F*).

Next, we used a random forest classifier to define biomarkers in the gut microbiota that distinguish EA and W subjects (*Figure 1—figure supplement 1B-D*). Classifiers employing ASV data and PhILR transformed phylogenetic nodes were trained using leave-one-out cross-validation. *B. obeum* (ASV1) was the top contributor to the resulting classifier, followed by *Anaerostipes hadrus* (ASV45) and then *Streptococcus parasanguinis* (ASV110) (*Figure 1—figure supplement 1B*). Both classifiers demonstrated the ability to distinguish between ethnic groups, with PhILR transformed phylogenetic nodes achieving a higher area under the curve compared to ASVs (*Figure 1—figure supplement 1C,D*). The majority (18/23) of the top ASVs identified by our classifier were also significantly different between ethnicities (*Figure 1—figure supplement 1E*).

Metagenomic sequencing provided independent confirmation of differences in the gut microbiome between ethnicities (*Supplementary file 1B*, n=21 EA, 24 W subjects). Consistent with our 16S-seq analysis, we detected a difference in the gut microbiomes between ethnicities based upon metagenomic species abundances (p=0.003, $R^2$=0.047, ADONIS, *Figure 2A*) and gene families (p=0.029, $R^2$=0.036, ADONIS). Ethnicity explained more variation in species abundances than a selection of demographic, laboratory, lifestyle, and metabolic metadata (*Figure 2B*). Visualization of diversity and species assignments within each phylum revealed marked variation in the magnitude and direction of change between individuals of a given ethnicity (*Figure 2C*). Genera that were found to be significantly different between ethnicities in our metagenomic data included *Akkermansia* and an unspecified *Erysipelotrichaceae* genera (*Figure 2D*) elevated in W individuals. Four bacterial species were significantly different between ethnicities in our metagenomic data: W individuals had higher levels of *A. muciniphila*, *Bacteroidales bacterium ph8*, and *Roseburia hominis*, and lower levels of *Ruminococcus gnavus*, compared to EA individuals (*Figure 2E*).

Next, we used nuclear magnetic resonance (NMR)-based stool metabolomics to gain insight into the potential functional consequences of ethnicity-associated differences in the human gut microbiome (*Supplementary file 1B*, n=10 subjects/ethnicity). Metabolite profiles were more strongly associated with ethnicity (p=0.008, $R^2$=0.128, ADONIS; *Figure 3A*) than community structure ($R^2$=0.029–0.055, ADONIS; *Supplementary file 1C*) or gene abundance (p=0.029, $R^2$=0.036, ADONIS). Feature annotations revealed elevated levels of the branched-chain amino acid (BCAA) valine and the short-chain fatty acids (SCFAs) acetate and propionate in EA subjects (*Figure 3B* and *Supplementary file 1D*).

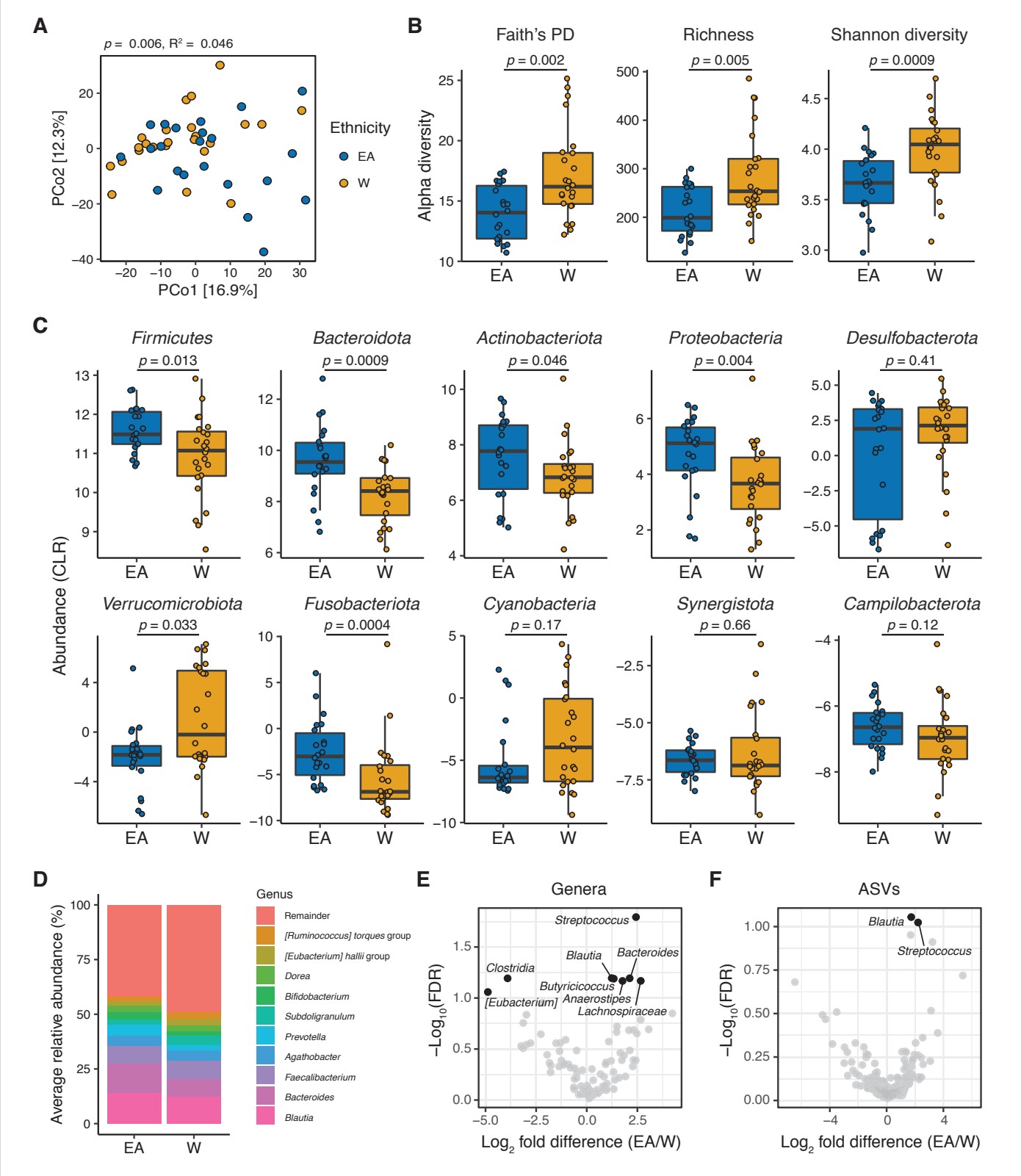

**Figure 1.** The gut microbiota is distinct between East Asian (EA) and White (W) subjects living in the Bay Area. (**A–C**) Each point represents a single individual's gut microbiota based upon 16S-seq. (**A**) Principal coordinate analysis of PhILR Euclidean distances reveals significant separation between ethnic groups (ADONIS test values shown). Additional distance calculations for complementary distance matrix calculations are shown in **Supplementary file 1C**. (**B**) Calculations of alpha diversity between EA and W subjects. p-values determined using Wilcoxon rank-sum tests. (**C**) CLR

*Figure 1 continued on next page*

*Figure 1 continued*

abundances of all bacterial phyla between EA and W subjects. p-values determined using Wilcoxon rank-sum tests. (**D**) Stacked bar plots showing the average percent relative abundances at the genus level for EA and W subjects, respectively. The most abundant taxa are shown as differently colored bars, with lower abundance taxa grouped as a single bar (Remainder). (**E, F**) Volcano plot of ALDEx2 differential abundance testing on (**E**) genera and (**F**) ASVs detected by 16S-seq in the gut microbiotas of EA versus W individuals. Significantly different (FDR<0.1) features are highlighted in black and labeled by genus or the most specific taxonomic assignment. (**A–F**) n=22 EA and n=24 W individuals. ASV, amplicon sequence variant; FDR, false discovery rate.

The online version of this article includes the following figure supplement(s) for figure 1:

**Figure supplement 1.** The gut microbiota can be used to predict ethnicity.

In contrast, proline, formate, alanine, xanthine, and hypoxanthine were found at higher levels in W subjects (*Figure 3B*). To assess the statistical significance and reproducibility of these trends, we used targeted gas chromatography mass spectrometry (GC-MS) and UPLC-MS/MS to quantify a panel of BCAAs, SCFAs, and bile acids (*Supplementary file 1E*). Confirming our NMR data, EA subjects had significantly higher levels of stool acetate (*Figure 3C*) and propionate (*Figure 3D*); however, we did not detect any significant differences in BCAAs or bile acids (*Figure 3—figure supplement 1*). Isobutyrate (which was not detected by NMR) was also significantly higher in EA subjects (*Figure 3E*). In agreement with these metabolite levels, a targeted re-analysis of our metagenomic data revealed a

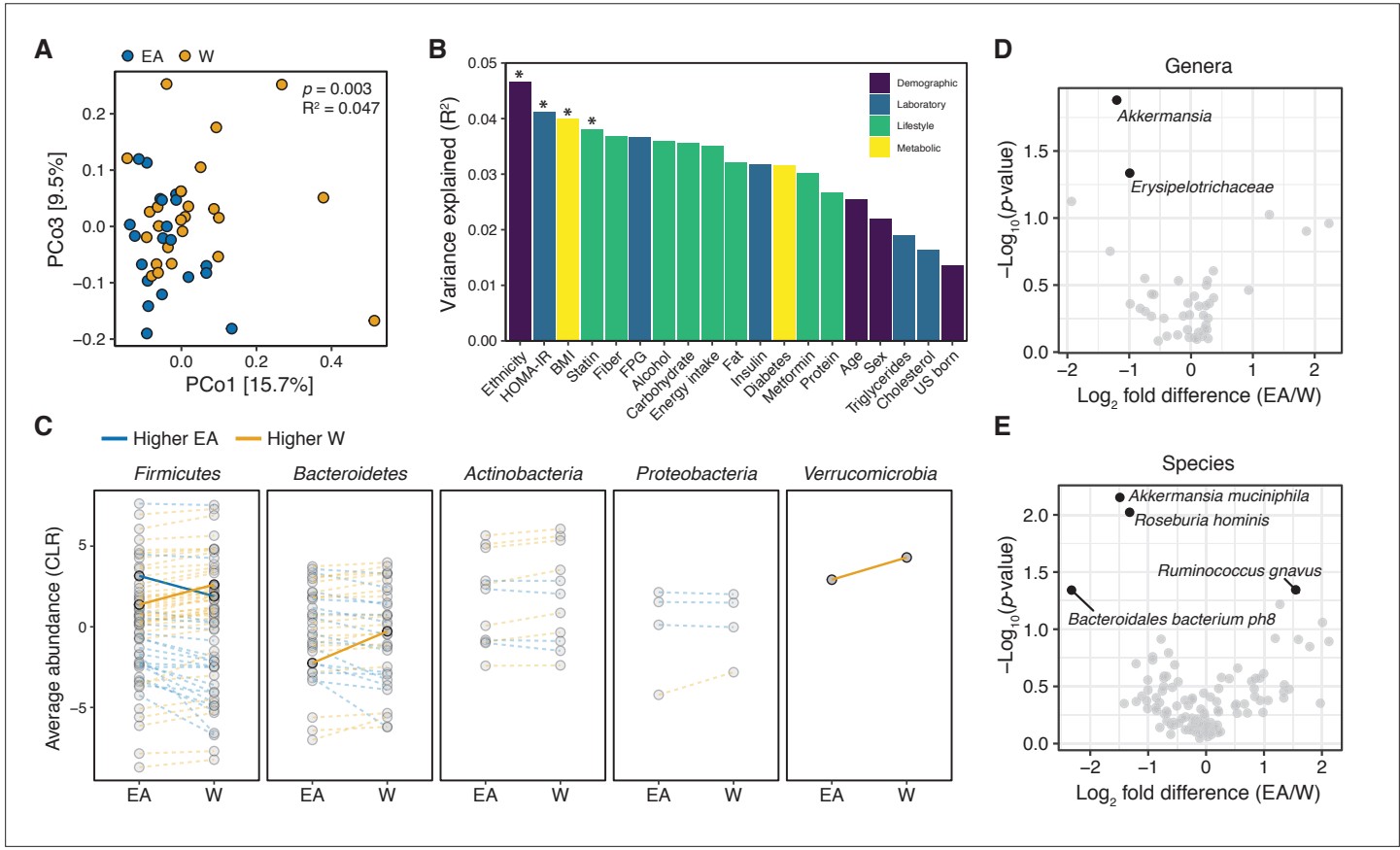

**Figure 2.** Metagenomic sequencing corroborates differences in the gut microbiota between ethnicities. (**A**) Principal coordinate analysis of Bray-Curtis distances reveals significant separation between ethnic groups (ADONIS test values shown). Each point represents a single individual's gut microbiota based upon shotgun sequencing. (**B**) PERMANOVA calculations for metadata variables on the x-axis with relation to variance in shotgun transformed species data with resulting effect size plotted on the y-axis. Variables are colored by metadata type (see inset; *p<0.05, ADONIS). (**C**) Each point represents the average relative abundance for a given species within each ethnic group, connected with a line that is colored by the ethnic group with higher mean abundance of each species: EA (blue) and W (orange). Solid lines highlight four bacterial species that are significantly different between ethnicity (p<0.05, ALDEx2, also shown in (E)). (**D, E**) Volcano plot of ALDEx2 differential abundance testing on (**D**) genera and (**E**) species level shotgun data. Significantly different (p<0.05) features are highlighted in black and labeled by the most specific taxonomic assignment. (**A–E**) n=21 EA and n=24 W individuals. Data reflects metagenomic sequencing. EA, East Asian; W, White.

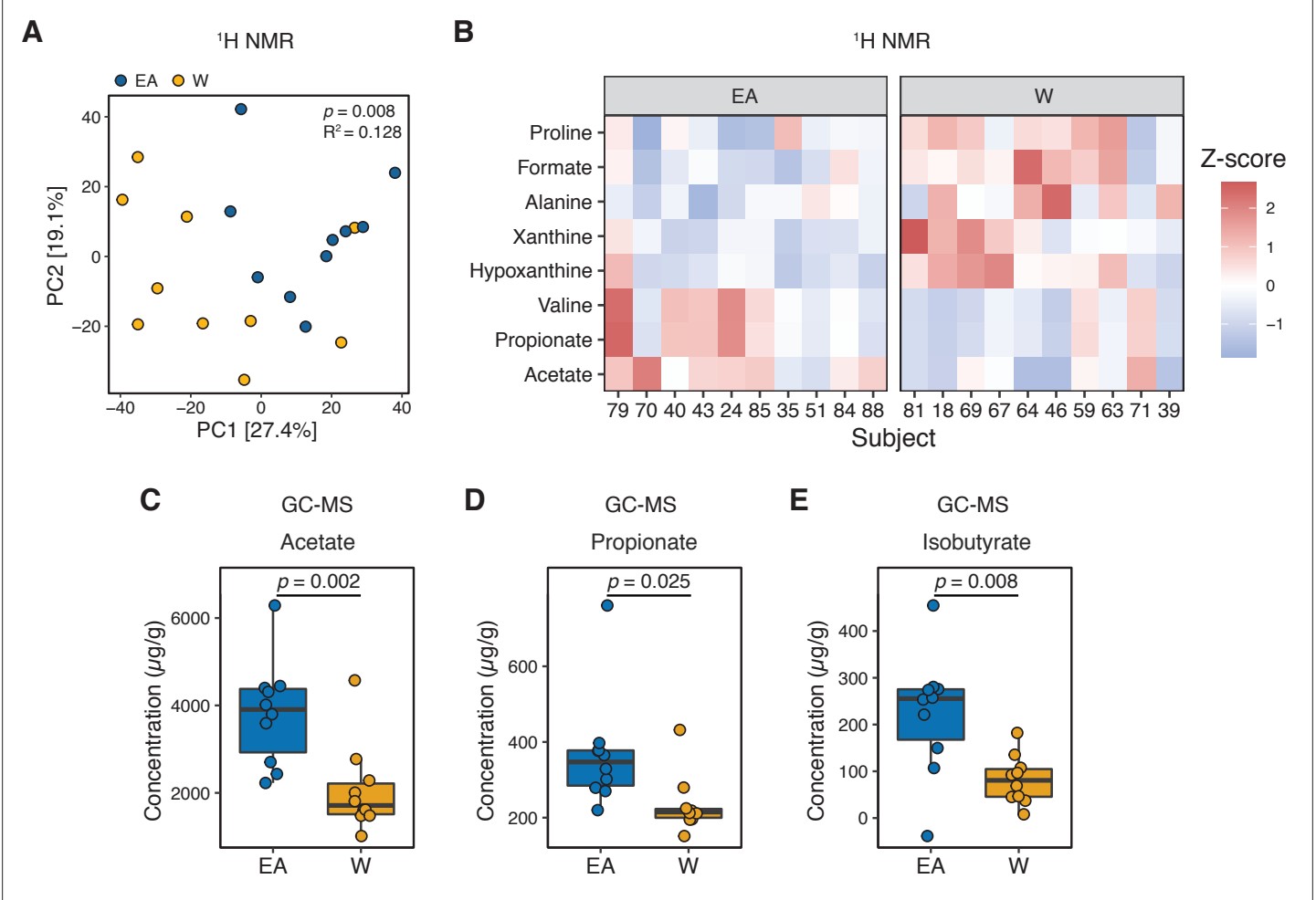

**Figure 3.** Metabolomics and targeted metabolite profiling highlight significant differences in bacterial fermentation end-products between ethnicities. (**A**) Global profiling of the stool metabolome by proton nuclear magnetic resonance (¹H NMR) revealed a significant separation in metabolomic profiles between EA and W individuals (ADONIS test values shown). (**B**) Representative stool metabolites contributing to the separation of stool metabolomic profiles between EA and W individuals (p<0.05, Wilcoxon rank-sum test). (**C–E**) Gas chromatography-mass spectrometry analysis of short-chain fatty acids (SCFAs) revealed significantly higher concentrations of acetate (**C**), propionate (**D**), and isobutyrate (**E**) in the stool samples of EA compared to W individuals. p-values determined using Wilcoxon rank-sum tests. (**A–E**) n=10 EA and n=10 W individuals. EA, East Asian W, White.

The online version of this article includes the following figure supplement(s) for figure 3:

**Figure supplement 1.** Stool concentrations of branched-chain amino acids (BCAAs) and bile acids are comparable between East Asian (EA) and White (W) subjects.

significant enrichment in two SCFA-related pathways: 'pyruvate fermentation to butanoate' (p=0.023, fold-difference=2.216) and 'superpathway of *Clostridium acetobutylicum* acidogenic fermentation' (p=0.023, fold-difference=2.182).

Consistent with prior work (*Le Chatelier et al., 2013*; *Turnbaugh et al., 2008*), we found that gut bacterial richness in W individuals was significantly associated with both BMI (*Figure 4A*) and body fat percentage (*Figure 4B*). Remarkably, these associations were undetectable in EA subjects (*Figure 4A and B*) even when other metrics of bacterial diversity were used (*Figure 4—figure supplement 1*),with the single exception of a negative correlation between Shannon diversity and BMI in EA subjects (*Figure 4—figure supplement 1C*). Re-analysis of our data separating lean and obese individuals revealed that the previously observed differences between ethnic groups were driven by lean individuals. Compared to lean EA individuals, lean W subjects had significantly higher bacterial diversity (*Figure 4C*) and more marked differences in gut microbial community structure (p=0.0003, R²=0.122, ADONIS; *Figure 4D*) and metabolite profiles (p=0.010, R²=0.293, ADONIS; *Figure 4E*). By

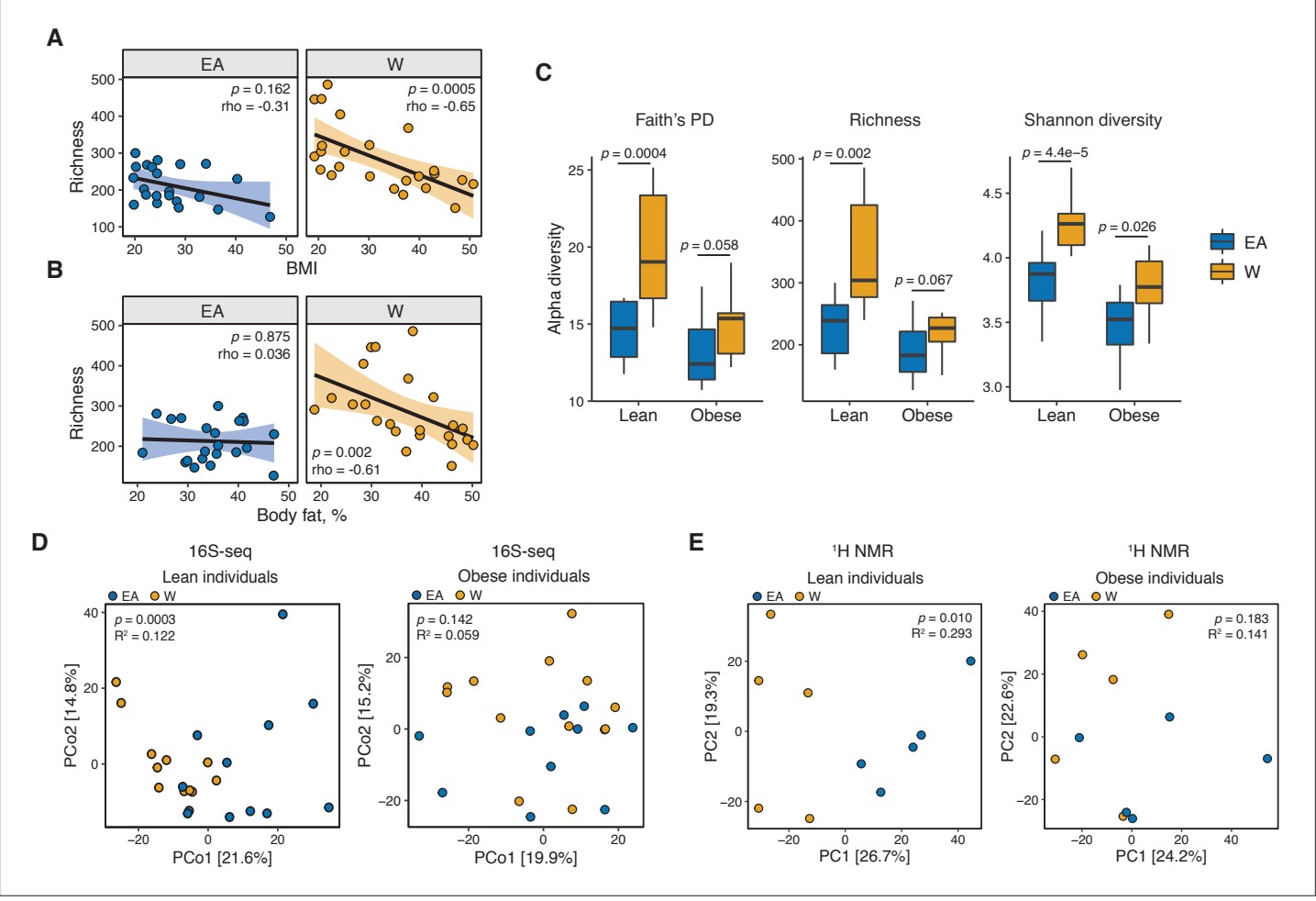

**Figure 4.** Ethnicity-associated differences in gut microbial diversity and community structure are more pronounced in lean individuals. (**A, B**) Bacterial richness is negatively correlated with (**A**) BMI and (**B**) percent body fat in W but not EA individuals (Spearman rank correlation coefficients and p-values are shown for each graph). (**C**) Microbial diversity metrics are more distinct between ethnic groups in lean relative to obese individuals. p-values determined using Wilcoxon rank-sum tests. (**D**) Principal coordinate analysis of PhILR Euclidean distances reveals significant separation between the gut microbiotas of EA and W lean individuals, with no separation in obese subjects (ADONIS test values shown). (**A–D**) n=12 EA lean, 10 EA obese, 11 W lean, and 13 W obese individuals. Data reflects 16S-seq. (**E**) Global profiling of the stool metabolome by proton nuclear magnetic resonance ($^1$H NMR) stratified by lean and obese individuals reveals a significant difference in the metabolomic profiles of lean EA and W individuals that is not detectable in obese individuals (ADONIS test values shown; n=5 individuals/group). EA, East Asian; W, White.

The online version of this article includes the following figure supplement(s) for figure 4:

**Figure supplement 1.** Microbial diversity metrics are consistently and negatively correlated with metabolic parameters in White individuals.

contrast, obese W versus EA individuals were not different across any of these metrics (*Figure 4C–E*), except for lower Shannon diversity in obese EA compared to W individuals (*Figure 4C*). We also detected differences in the gut microbiotas of lean EA and W individuals at the phylum (*Figure 5A*) and genus (*Figure 5B*) levels that were largely consistent with our original analysis of the full data set (*Figure 1C and E*). More modest differences in the gut microbiota between ethnicities were observed in obese subjects (*Figure 5A and C*).

Next, we sought to understand the potential drivers of differences in the gut microbiome between ethnic groups in lean individuals within the IDEO cohort. Consistent with prior studies (*Falony et al., 2016*), PERMANOVA analysis of our full 16S-seq data set revealed that diabetes (*Forslund et al., 2015*), age (*Ghosh et al., 2020*), metformin use (*Wu et al., 2017*), and statin intake (*Vieira-Silva et al., 2020*) were significantly associated with variance in the PhILR Euclidean distances (*Figure 6—figure supplement 1*). Metagenomic sequencing of the IDEO cohort with subsequent PERMANOVA analysis confirmed significant associations with ethnicity and statin use, while also highlighting significant

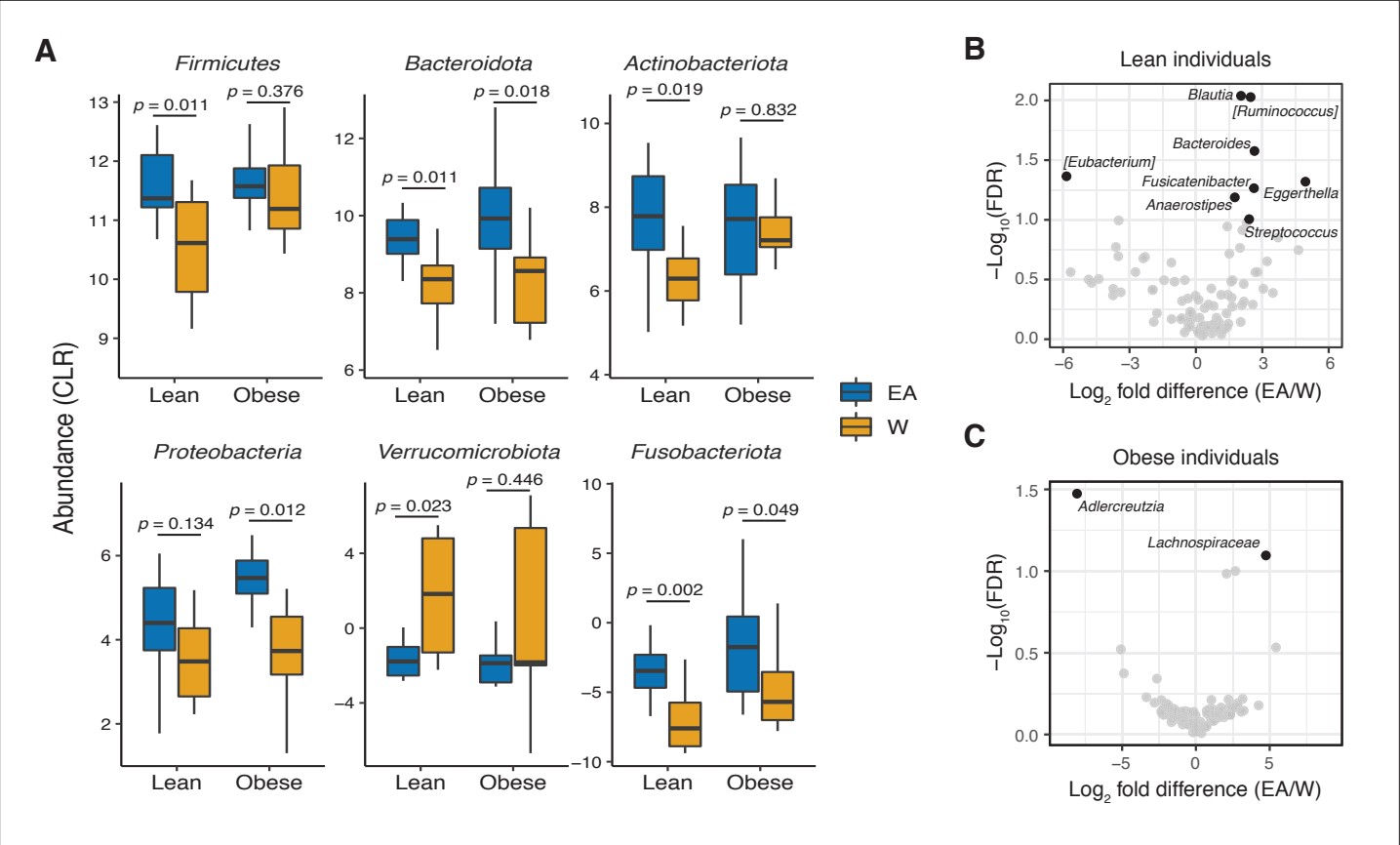

**Figure 5.** Ethnicity-associated bacterial taxa in lean and obese individuals. (**A**) 5/6 phyla that were differentially abundant between ethnicities (see *Figure 1C*) were also significantly different between lean EA and W individuals. Three phyla were significantly different between obese EA and W individuals (p<0.05, Wilcoxon rank-sum test). (**B, C**) Volcano plot of ALDEx2 differential abundance testing on genera in stool microbiotas of lean (**B**) and obese (**C**) EA versus W individuals, with significantly different genera highlighted (FDR<0.1, ALDEx2). (**A–C**) n=12 EA lean, 10 EA obese, 11 W lean, and 13 W obese individuals. Data reflects 16S-seq. EA, East Asian; FDR, false discovery rate; W, White.

associations with HOMA-IR and BMI (*Figure 2B*), consistent with prior reports (*Liu et al., 2017*; *Zouiouich et al., 2021*). While several factors linked to body composition were different between obese EA and W subjects using a nominal p-value, only triglyceride levels were significantly different between lean EA and W subjects and this trend did not survive multiple testing correction (*Supplementary file 1A*). Although everyone in the cohort was recruited from the San Francisco Bay Area, birth location varied widely (*Figure 6—figure supplement 2*). There was no significant difference in the proportion of subjects born in the United States between ethnicities (75% W, 54.5% EA; p=0.15, Pearson's $\chi^2$ test). There was also no significant difference in the geographical distance between birth location and San Francisco [W median 2,318 (2.2–6,906) miles; EA median 1,986 (2.2–6,906) miles; p=0.69, Wilcoxon rank-sum test] or the amount of time spent in the San Francisco Bay Area at the time of sampling [W median 270 (8.00–741) months; EA median 282.5 (8.50–777) months; p=0.42, Wilcoxon rank-sum test].

Surprisingly, we did not detect any significant differences in either short- (*Supplementary file 1F*) or long-term (*Supplementary file 1G*) dietary intake between ethnicities. Consistent with this, procrustes analysis did not reveal any significant associations between dietary intake and gut microbial community structure: procrustes p=0.280 (DHQIII) and p=0.080 (ASA24) relative to PhILR transformed 16S-seq ASV data. The Spearman Mantel statistic was also non-significant [r=0.0524, p=0.243 (DHQIII) and r=−0.0173, p=0.590 (ASA24)], relative to PhILR transformed 16S-seq ASV data. Despite the lack of an overall association between reported dietary intake and the gut microbiota, we were able to identify 12 ASVs and 7 metagenomic species associated with dietary intake in lean W individuals (*Figure 6—figure supplement 3A*). We also detected 20 significant species-level associations in

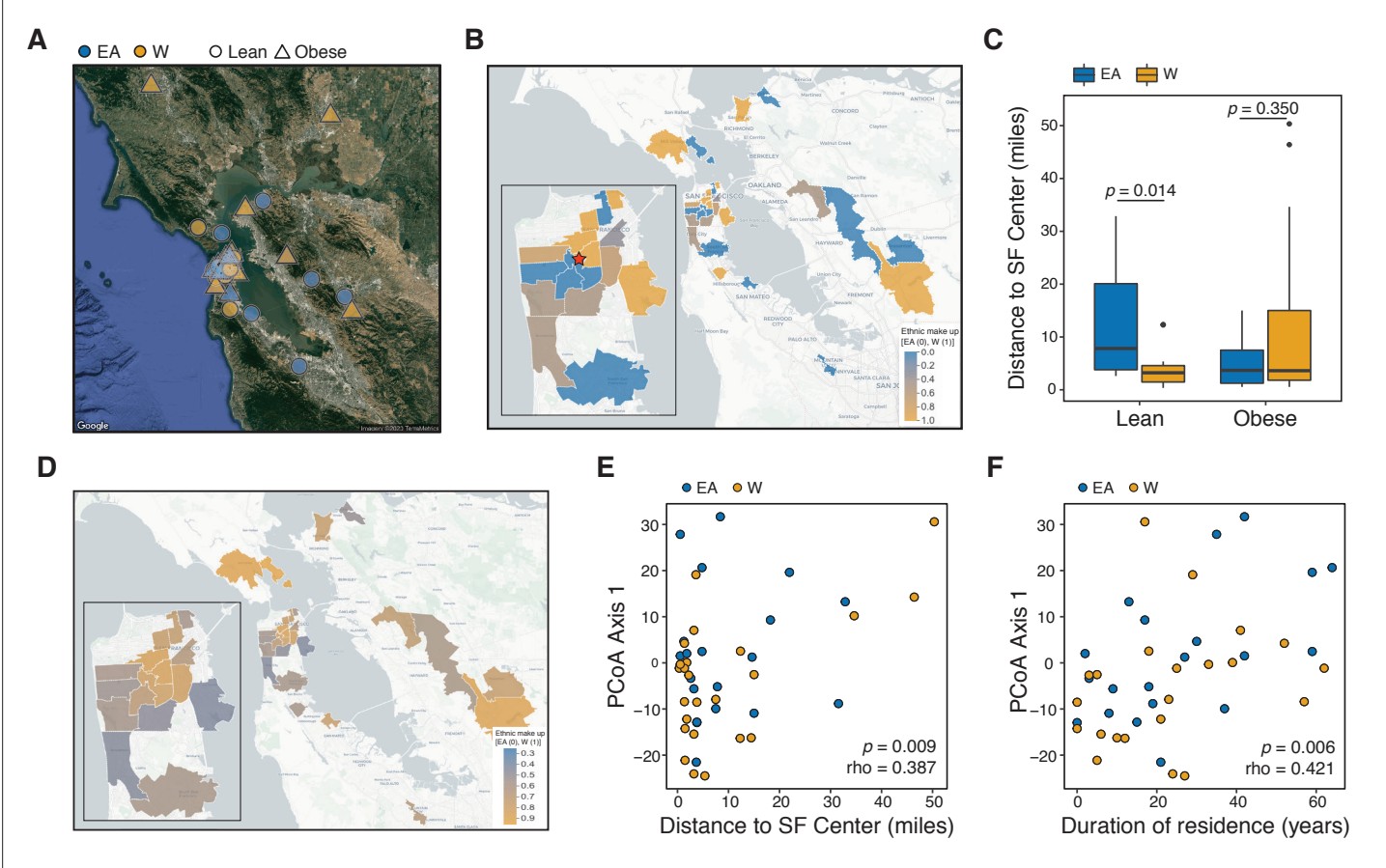

**Figure 6.** Ethnicity-associated differences in the gut microbiota of lean individuals correlate with geographic location. (**A**) Each symbol represents a subject's ZIP code. Symbols are colored by ethnicity with shape representing lean and obese subjects (n=44, data was unavailable for two subjects; *Supplementary file 1B*). (**B**) A subset of ZIP Code Tabulation Areas (ZCTAs) zoomed in to focus on San Francisco are colored by the proportion of each ethnicity (n=27 ZTCAs). The red star indicates a central point (latitude=37.7585102, longitude=−122.4539916) within San Francisco used for distances calculated in (**C**) . (**C**) Distance to the center of San Francisco, which is indicated by a star in (**B**), for IDEO subjects stratified by ethnicity and BMI (n=9–13 individuals/group, p-values indicate Wilcoxon rank-sum test). (**D**) US census data for EA and W residents in ZCTAs from (**B**) is displayed by ethnic make-up (a total of 489,117 W and 347,200 EA individuals in these areas). (**E, F**) PCoA principal coordinate axis 1 from PhILR Euclidean distances of the 16S-seq data is significantly correlated with (**E**) the distance of subject's ZIP code to the center of San Francisco and (**F**) the subject's duration of residence in the SF Bay Area (n=44 subjects; Spearman rank correlation). Data in (**E**) and (**F**) reflects 16S-seq.

The online version of this article includes the following figure supplement(s) for figure 6:

**Figure supplement 1.** Variables associated with variance in microbial 16S-seq data.

**Figure supplement 2.** Birth location of subjects.

**Figure supplement 3.** Identification of bacterial taxa associated with short-term dietary intake.

lean EA subjects (*Figure 6—figure supplement 3B*). There were no overlapping associations between ethnicities.

Given the marked variation in the gut microbiome at the continental scale (*Hehemann et al., 2010*; *Vangay et al., 2018*; *Yatsunenko et al., 2012*), we hypothesized that the observed differences in lean EA and W individuals may be influenced by a participant's current address at the time of sampling. Consistent with this hypothesis, we found clear trends in ethnic group composition across ZIP codes in the IDEO cohort (*Figure 6A and B*) that were mirrored by the 2018 US census data (Pearson r=0.52, p=0.026 for neighborhoods with greater than 50% W subjects; *Figure 6D*). Obese individuals from both ethnicities and lean W subjects tended to live closer to the center of San Francisco relative to lean EA subjects (*Figure 6C*). Distance between the current ZIP code and the center of San Francisco and duration of residency within San Francisco were both associated with gut microbial community structure (*Figure 6E and F*). The association between the current address and the gut microbiota was

robust to the central point used, as evidenced by using the Bay Bridge as the central reference point (p=0.008, rho=0.394, Spearman correlation).

Taken together, our results support the hypothesis that there are stable ethnicity-associated signatures within the gut microbiota of lean EA versus W individuals that are independent of diet. To experimentally test this hypothesis, we transplanted the gut microbiotas of two representative lean W and lean EA individuals into germ-free male C57BL/6J mice fed a low-fat, high-plant-polysaccharide (LFPP) diet (two independent experiments; per group n = 12 mice, 2 donors; per donor n=6 mice, 1 isolator; *Figure 7—figure supplement 1A,B*). The donors for this and the subsequent experiment were matched for their metabolic and other phenotypes to minimize potential confounding factors ( *Supplementary file 1H and I*). Despite maintaining the genetically identical recipient mice on the same autoclaved LFPP diet, we detected significant differences in gut microbial community structure (*Figure 7A*), bacterial richness (*Figure 7C*), and taxonomic abundance (*Figure 7D and E* and *Supplementary file 1J*) between the two ethnicity-specific recipient groups. These differences recapitulated key aspects of the gut microbiota observed in the IDEO cohort, including significantly lower bacterial richness (*Figure 7C*) and higher abundance of *Bacteroides* (*Figure 7D and E*) in recipient mice transplanted with microbiota from EA compared to W donors.

Next, we sought to assess the reproducibility of these findings across multiple donors and in the context of a distinctive dietary pressure. We fed 20 germ-free male mice a high-fat, high-sugar (HFHS) diet for 4 weeks prior to colonization with a gut microbiota from 1 of 5 W and 5 EA donors. Mice were maintained on the HFHS diet following colonization (per group n=10 mice, 5 donors; per donor n=2 mice, 1 cage; *Figure 7—figure supplement 1C*). This experiment replicated our original findings on the LFPP diet, including significantly altered gut microbial community structure between ethnicities (*Figure 7F*), significantly increased richness in mice receiving W donor microbiota (*Figure 7H*), and a trend toward higher levels of *Bacteroides* in mice receiving the gut microbiotas of EA donors (*Figure 7I and J*). Of note, the variance explained by ethnicity was lower in mice fed the HFHS diet ($R^2$=0.126) than the LFPP diet ($R^2$=0.384), potentially suggesting that in the context of human obesity, excessive fat and sugar consumption may serve to diminish the signal otherwise associated with ethnicity. As expected (*Nayak et al., 2021*; *Turnbaugh et al., 2009*; *Walter et al., 2020*), the input donor microbiota was distinct from that of the recipient mice (*Figure 7B and G*); however, there was no difference between ethnic groups in the efficiency of engraftment (*Figure 7—figure supplement 2*). In a pooled analysis of all gnotobiotic experiments accounting for one donor for multiple recipient mice, ethnicity and diet were both significantly associated with variations in the gut microbiota (*Figure 7—figure supplement 3*), consistent with the extensive published data demonstrating the rapid and reproducible impact of an HFHS diet on the mouse and human gut microbiota (*Bisanz et al., 2019*).

Finally, mice transplanted with gut microbiomes of EA and W individuals displayed differences in body composition. LFPP fed mice that received W donor microbiota had significantly increased adiposity in conjunction with decreased lean mass, relative to LFPP fed mice that received the EA donor microbiota (*Figure 8A–C*). Although these trends were mirrored in recipient mice that fed the HFHS diet (*Figure 8E–G*), they did not reach statistical significance. There were no significant differences in glucose tolerance in either experiment (*Figure 8D and H*). Taken together, these results suggest that dietary input may mask the metabolic consequences of ethnicity-associated differences in the gut microbiota.

## Discussion

Despite the potential for immigration to erase some of the geographically specific aspects of gut microbiome structure (*Vangay et al., 2018*), our study suggests that even in a given geographic location, there remain stable long-lasting microbial signatures of ethnicity, as revealed here for W and EA residents of the San Francisco Bay Area. The mechanisms responsible remain to be elucidated. In lean individuals within the IDEO cohort, these differences appear to be independent of immigration status, host phenotype, or dietary intake. Our experiments using inbred germ-free mice support the stability of ethnicity-associated differences in the gut microbiota on both the LFPP and HFHS diets, while also demonstrating that variations in host genetics are not necessary to maintain these signatures, at least over short timescales. Even though we conducted multiple experiments and recipient mice from the same donor generally mapped together, differences between the human donor and recipient mouse

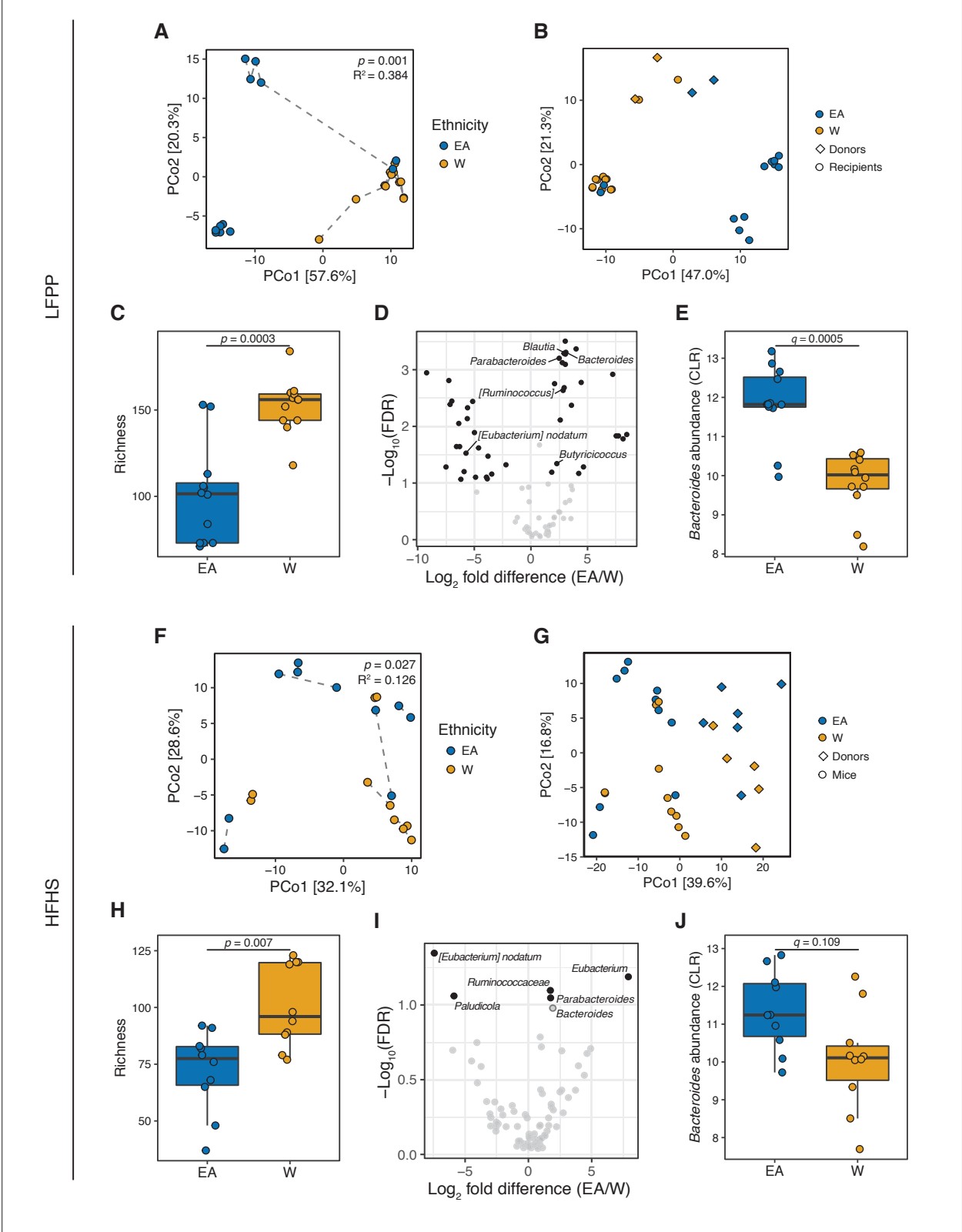

**Figure 7.** Differences in the human gut microbiota between ethnicities are maintained following transplantation to germ-free mice. (**A, F**) Principal coordinate analysis of PhILR Euclidean distances of stool from germ-free recipient mice transplanted with stool microbial communities from lean EA or W donors and fed either an LFPP (**A**, combined results from two independent experiments; n=12 recipient mice per group) or HFHS (**F**, n=10 recipient mice per group) diet. Significance was assessed by ADONIS. Germ-free mice receiving the same donor sample are connected by a dashed

*Figure 7 continued on next page*

*Figure 7 continued*

line. Experimental designs are shown in *Figure 7—figure supplement 1*. (**B, G**) Principal coordinate analysis of PhILR Euclidean distances comparing donor input slurry (diamonds) and stool from recipient mice (circles) in the combined LFPP experiments (**B**, n=4 donors, 24 recipients) and HFHS experiment (**G**, n=10 donors, 20 recipients; for one donor sample, two separate slurries were prepared to inoculate the recipient mice on separate days due to constraints on germ-free mice availability, resulting in 11 diamonds on the plot). See also donor metadata in *Supplementary file 1B,H*. (**C, H**) Bacterial richness is significantly higher in mice who received stool samples from W donors compared to EA donors on both the LFPP (**C**) and HFHS (**H**) diets. p-values determined using Wilcoxon rank-sum tests. (**D, I**) Volcano plot of ALDEx2 differential abundance testing on genera in the stool microbiomes between transplant groups. The x-axis represents the fold difference between EA (numerator) and W (denominator) subjects. The y-axis is proportional to the false discovery rate (FDR). Black dots indicate significantly different genera (FDR<0.1). *Bacteroides* and *Parabacteroides* (labeled in the volcano plots) are more abundant in mice that received stool samples from EA compared to W donors on both the LFPP (**D**) and HFHS (**I**) diets. See also *Supplementary file 1J* for the full list of significant genera. (**E, J**) Abundance of the *Bacteroides* genus in mice fed the LFPP (**E**) and HFHS (**J**) diets (ALDEx2 FDR shown). Data reflects 16S-seq. EA, East Asian; HFHS, high-fat, high-sugar; LFPP, low-fat, high-plant-polysaccharide; W, White.

The online version of this article includes the following figure supplement(s) for figure 7:

**Figure supplement 1.** Experimental designs for gnotobiotic experiments.

**Figure supplement 2.** Engraftment efficiency is comparable between donor groups.

**Figure supplement 3.** Combined analysis of recipient mice reveals significant associations with donor ethnicity and recipient diet.

microbiotas inherent to gnotobiotic transplantation warrant further investigation, as do differences in the stability of the gut microbiotas of male versus female donors.

Our data also supports a potential role for geographic location of residence in reinforcing differences in the gut microbiota between ethnic groups. The specific reasons why current location would

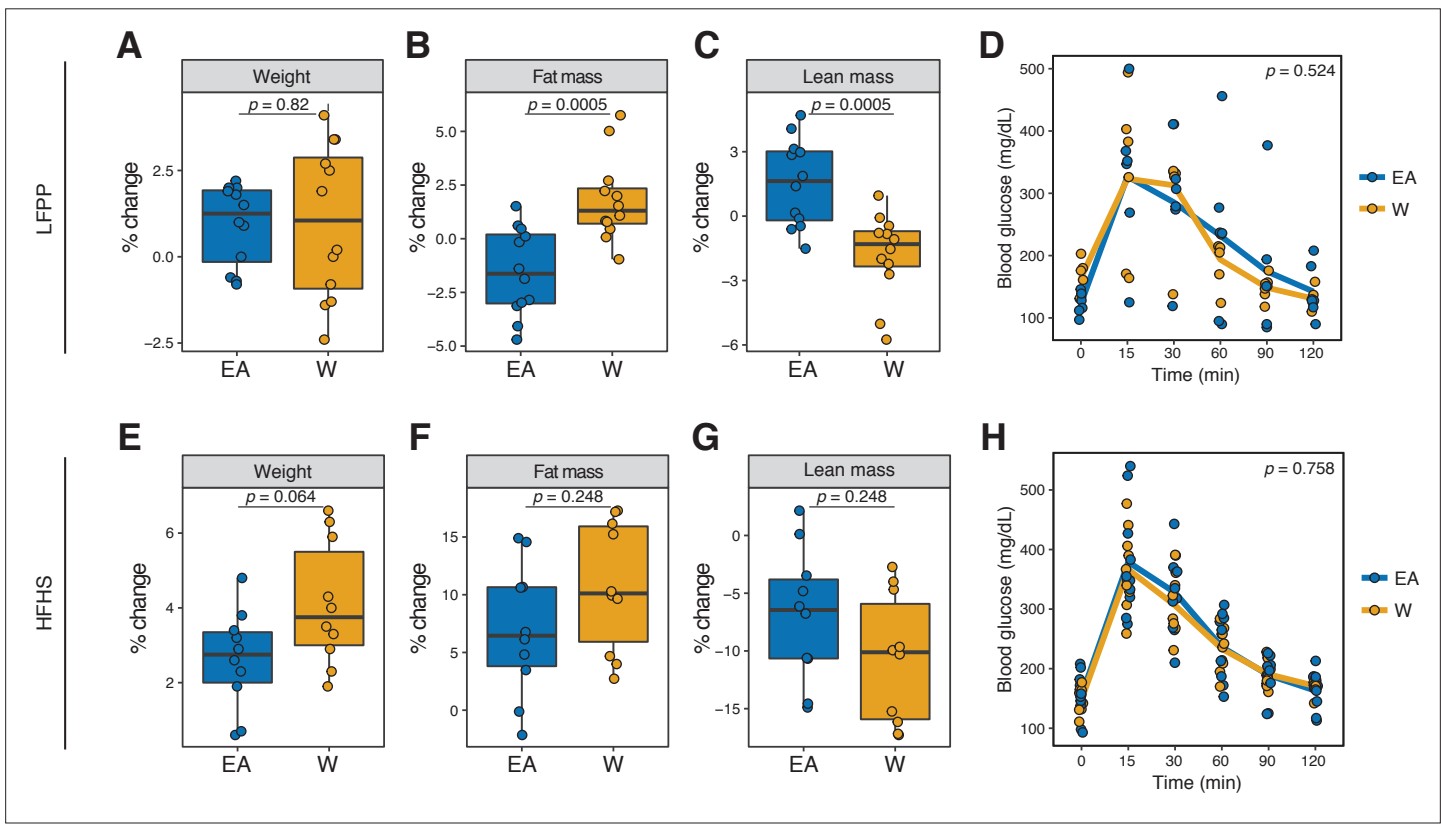

**Figure 8.** Microbiome transplantation of samples from EA and W individuals differentially affects the body composition of genetically identical recipient mice. (**A–C, E–G**) Percent change in body weight (**A, E**), fat mass (**B, F**), and lean mass (**C, G**) relative to baseline are shown on the LFPP (**A–C**) and HFHS (**E–G**) diets. p-values determined using Wilcoxon rank-sum tests. (**D, H**) Glucose tolerance test results were not significantly different between groups on either diet. p-values determined using linear mixed-effects models with mouse as a random effect. (**A–C**) n=12 recipient mice per group (combined data from two independent experiments). (**D**) n=6 recipient mice per group from a single experiment. (**E–H**) n=10 recipient mice per group. Experimental designs are shown in *Figure 7—figure supplement 1* and donor phenotypic data is in *Supplementary file 1H*. EA, East Asian; HFHS, high-fat, high-sugar; LFPP, low-fat, high-plant-polysaccharide; W, White.

matter to the gut microbiota remain unclear. Current location may reflect subtle differences in dietary intake (e.g., ethnic foods, food sources, or phytochemical contents) that are hard to capture using the validated nutritional surveys employed here (*Garduño-Diaz et al., 2014*). Alternative hypotheses include biogeographical patterns in microbial dispersion (*Martiny et al., 2006*) or a role for socioeconomic factors, which are correlated with neighborhood (*Kakar et al., 2018*).

Surprisingly, our findings demonstrate that ethnicity-associated differences in the gut microbiota are stronger in lean individuals. Obese individuals did not exhibit as clear a difference in the gut microbiota between ethnic groups, either suggesting that established obesity or its associated dietary patterns can overwrite long-lasting microbial signatures. Alternatively, there could be a shared ethnicity-independent microbiome type that predisposes individuals to obesity. Studies in other disease areas (e.g., inflammatory bowel disease and cancer) with similar multi-ethnic cohorts are essential to test the generalizability of these findings and to generate hypotheses as to their mechanistic underpinnings.

Our results in humans and mouse models support the broad potential for downstream consequences of ethnicity-associated differences in the gut microbiome for metabolic syndrome and potentially other disease areas. However, the causal relationships and how they can be understood in the context of the broader differences in host phenotype between ethnicities require further study. While these data are consistent with our general hypothesis that ethnicity-associated differences in the gut microbiome are a source of differences in host metabolic disease risk, we were surprised by both the nature of the microbiome shifts and their directionality. Based upon observations in the IDEO (*Alba et al., 2018*) and other cohorts (*Gu et al., 2006*; *Zheng et al., 2011*), we anticipated that the gut microbiomes of lean EA individuals would promote obesity or other features of metabolic syndrome. In humans, we did find multiple signals that have been previously linked to obesity and its associated metabolic diseases in EA individuals, including increased Firmicutes (*Basolo et al., 2020*; *Bisanz et al., 2019*), decreased *A. muciniphila* (*Depommier et al., 2019*; *Plovier et al., 2017*), decreased diversity (*Turnbaugh et al., 2008*), and increased acetate (*Perry et al., 2016*; *Turnbaugh et al., 2006*). Yet EA subjects also had higher levels of *Bacteroidota* and *Bacteroides*, which have been linked to improved metabolic health (*Johnson et al., 2017*). More importantly, our microbiome transplantations demonstrated that the recipients of the lean EA gut microbiome had less body fat despite consuming the same diet. These seemingly contradictory findings may suggest that the recipient mice lost some of the microbial features of ethnicity relevant to host metabolic disease or alternatively that the microbiome acts in a beneficial manner to counteract other ethnicity-associated factors driving disease.

EA subjects also had elevated levels of the SCFAs propionate and isobutyrate. The consequences of elevated intestinal propionate levels are unclear given the seemingly conflicting evidence in the literature that propionate may either exacerbate (*Tirosh et al., 2019*) or protect from (*Lu et al., 2016*) aspects of metabolic syndrome. Clinical data suggests that circulating propionate may be more relevant for disease than fecal levels (*Müller et al., 2019*), emphasizing the importance of considering both the specific microbial metabolites produced, their intestinal absorption, and their distribution throughout the body. Isobutyrate is even less well-characterized, with prior links to dietary intake (*Berding and Donovan, 2018*) but no association with obesity (*Kim et al., 2019*). Unlike SCFAs, we did not identify consistent differences in BCAAs, potentially due to differences in both extraction and standardization techniques inherent to GC-MS and NMR analysis (*Cai et al., 2016*; *Lynch and Adams, 2014*; *Qin et al., 2012*).

There are multiple limitations of this study. Due to the investment of resources into ensuring a high level of phenotypic information on each cohort member coupled to the restricted geographical catchment area, the IDEO cohort was relatively small at the time of this analysis (n=46 individuals). The current study only focused on two of the major ethnicities in the San Francisco Bay Area. As IDEO continues to expand and diversify its membership, we hope to study participants from other ethnic groups. Stool samples were collected at a single time point and analyzed in a cross-sectional manner. While we used validated tools from the field of nutrition to monitor dietary intake, we cannot fully exclude subtle dietary differences between ethnicities (*Johnson et al., 2019*), which could be interrogated through controlled feeding studies (*Basolo et al., 2020*). Our mouse experiments were all performed in wild-type adult males. The use of a microbiome-dependent transgenic mouse model of diabetes (*Brown et al., 2016*) would be useful to test the effects of inter-ethnic differences in the microbiome on insulin and glucose tolerance. Additional experiments are warranted using the same donor inocula to colonize germ-free mice prior to concomitant feeding of multiple diets, allowing a more explicit test of the hypothesis that diet can disrupt ethnicity-associated microbial signatures.

These studies, coupled to controlled experimentation with individual strains or more complex synthetic communities, would help to elucidate the mechanisms responsible for ethnicity-associated changes in host physiology and their relevance to disease.

## Conclusions

Our results support the utility of considering ethnicity as a covariate in microbiome studies, due to the ability to detect signals that are difficult to capture by more specific metadata such as individual dietary intake values. On the other hand, these findings raise the importance of dissecting the sociological and biological components of ethnicity with the goal of identifying factors that shape the gut microbiota, either alone or in combination. This emerging area of microbiome research is just one component in the broader efforts to explore the boundaries and mechanistic underpinning of ethnicity with respect to multiple ethnic groups. The IDEO cohort provides a valuable research tool to conduct prospective longitudinal and intervention studies examining diabetes in diverse participants. More broadly, IDEO provides a framework to approach other disease states where self-identified race or ethnicity are thought to contribute to health outcomes related to the microbiome, including the use of gnotobiotic mouse models to examine the specific role of microbial communities in contributing to phenotypes linked to ethnicity. By understanding the biological features that drive differences between ethnic groups, we may be able to achieve similar health outcomes and to support more precise therapies informed by a broader appreciation of both microbial and human diversity.

# Materials and methods

**Key resources table**

| Reagent type (species) or resource | Designation | Source or reference | Identifiers | Additional information |
|---|---|---|---|---|
| Biological sample (*Homo sapiens*) | Stool | This paper | | n=46 subjects (22 EA, 24 W) |
| Commercial assay or kit | Wizard SV 96 Genomic DNA kit | Promega | Cat #A2370 | |
| Commercial assay or kit | ZymoBIOMICS 96 MagBead DNA Kit | Zymo Research | Cat #D4302 | |
| Software, algorithm | R | CRAN | v3.5.3, v4.0.2 | https://cran.r-project.org |
| Software, algorithm | QIIME2 | *Bolyen et al., 2019* | v2020.2 | http://qiime2.org |
| Software, algorithm | DADA2 | *Callahan et al., 2016* | | http://benjjneb.github.io/dada2 |
| Software, algorithm | MicrobeR | *Bisanz, 2017* | v0.3.2 | https://github.com/jbisanz/MicrobeR |
| Software, algorithm | qiime2R | *Bisanz, 2018* | v0.99.34 | http://github.com/jbisanz/qiime2R |
| Software, algorithm | MetaPhlAn2 | *Truong et al., 2015* | v2.7.7 | http://huttenhower.sph.harvard.edu/metaphlan2 |
| Software, algorithm | Vegan | *Oksanen et al., 2013* | v2.5–6 | https://CRAN.R-project.org/package=vegan |
| Software, algorithm | APE | *Paradis and Schliep, 2019* | v5.3 | http://ape-package.ird.fr |
| Software, algorithm | Picante | *Kembel et al., 2010* | v1.8.1 | http://github.com/skembel/picante |
| Software, algorithm | PhILR | *Silverman et al., 2017* | v1.12.0 | http://github.com/jsilve24/philr |
| Software, algorithm | ALDEx2 | *Fernandes et al., 2013* | v1.18.0 | http://github.com/ggloor/ALDEx_bioc |
| Software, algorithm | GGMaps | *Kahle and Wickham, 2013* | v3.0.0.902 | https://github.com/dkahle/ggmap |
| Software, algorithm | Open Street Maps | https://www.openstreetmap.org | | https://wiki.openstreetmap.org/wiki/Main_Page |
| Software, algorithm | Leaflet | https://www.openstreetmap.org | v1.6.0 | http://rstudio.github.io/leaflet |
| Software, algorithm | Imap | *Wallace, 2012* | v1.32 | https://rdrr.io/cran/Imap |

*Continued on next page*

*Continued*

| Reagent type (species) or resource | Designation | Source or reference | Identifiers | Additional information |
|---|---|---|---|---|
| Strain, strain background (*Mus musculus*) | C57BL/6J mice, germ-free | UCSF Gnotobiotics Core | | |
| Other | 2018 US Census data | http://data.census.gov | | Table B02001: Race |

## Human subjects

The IDEO cohort was established to explore the pathogenesis of obesity and metabolic diseases in highly vulnerable segments of the population. It includes men and women of multiple ethnicities recruited from the general medicine, endocrinology, diabetes, general surgery, and bariatric surgery clinics at the University of California San Francisco (UCSF) and Zuckerberg San Francisco General Hospital and by public advertisements throughout the local San Francisco Bay Area. All study participants were part of the IDEO cohort, which has been previously described (*Alba et al., 2018*; *Oguri et al., 2020*). Briefly, IDEO consists of 25–65-year-old men and women of multiple ethnicities and across a wide BMI range (18.5–52 kg/m$^2$) living in the San Francisco Bay Area. Using IDEO, we recruited both lean and obese W and EA individuals into this study based on World Health Organization cutoffs: W/EA BMI≤24.9 kg/m$^2$ (lean); W BMI≥30 kg/m$^2$ (obese); and EA BMI≥27.5 kg/m$^2$ (obese) (*Hsu et al., 2015*; *Jih et al., 2014*; *Expert Consultation, 2004*). To avoid bias toward non-English speaking participants, all documents including flyers, screening questionnaires, and consents were available in Cantonese and Mandarin. Potential participants completed screening questionnaires and exclusion criteria were assessed in more detail. These included acute or chronic infections, current medications with a recognized impact on the immune system, recent antibiotic use, current smoking, recent changes in weight, active liver disease or liver failure, chronic kidney disease (eGFR <30 ml/min/1.73 m$^2$), history of cancer and chemotherapy therapy within the past 5 years, psychiatric and neurological disorders, prior bariatric surgery, and weight >159 kg (the DXA scanner weight limit). Whereas exclusion criteria inherently lend bias toward healthy individuals, this is done to limit the confounding effects of a wide variety of chronic diseases and environmental exposures on the comparisons being made.

IDEO also limited bias by standardizing how individuals are asked to self-identify race/ethnicity. Individuals are asked to respond to two separate questions about ethnicity (e.g., 'are you of Hispanic, Latino, or Spanish origin?') and race ('What is your race?'). Hispanic/LatinX individuals were enrolled as part of a separate IDEO sub-study from the topic of this manuscript. Participants are also asked questions about their parents' race and ethnic background. Each participant consented to take part in the study, which was approved by the UCSF Committee on Human Research. We utilized demographic, medical, dietary, and lifestyle metadata on each participant that were part of their initial recruitment into IDEO, as previously reported (*Alba et al., 2018*; *Oguri et al., 2020*). Participants with Type 2 Diabetes (T2D) were classified in accordance with American Diabetes Association Standards of Medical Care guidelines (*American Diabetes Association, 2019*), defined by having glycated hemoglobin (HbA1c)≥6.5% or the combination of a prior diagnosis of T2D and the active use of an antidiabetic medication. For stool sample collection, participants took home or were mailed a stool sample collection kit and detailed instructions on how to collect the specimen. All samples were collected at home, stored at room temperature, and brought to the UCSF Clinical Research Center by the participants within 24 hr of defecation. Samples were aliquoted and stored at – 80°C .

## Anthropometric and body composition measurements

We leveraged host phenotypic and demographic data from IDEO, which was the focus of two previous studies (*Alba et al., 2018*; *Oguri et al., 2020*). For the convenience of the reader, we restate our methods here. Height and weight were measured using a standard stadiometer and scale, and BMI (kg/m$^2$) was calculated from two averaged measurements. Waist and hip circumferences (to the nearest 0.5 cm) were measured using a plastic tape meter at the level of the umbilicus and of the greater trochanters, respectively, and waist-to-hip ratio (WHR) was calculated. Blood pressure was measured with a standard mercury sphygmomanometer on the left arm after at least 10 min of rest. Mean values were determined from two independent measurements. Blood samples were collected after an overnight fast and analyzed for plasma glucose, insulin, serum total cholesterol, high-density

lipoprotein (HDL) cholesterol, and triglycerides. Low-density lipoprotein (LDL) cholesterol was estimated according to the Friedewald formula (*Friedewald et al., 1972*). Insulin resistance was estimated by the homeostatic model assessment of insulin resistance (HOMA-IR) index calculated from fasting glucose and insulin values (*Matthews et al., 1985*). Two obese subjects on insulin were included in the HOMA-IR analysis (1 EA, 1 W). Body composition of the subjects was estimated by Dual-Energy X-ray Absorptiometry (DEXA) using a Hologic Horizon/A scanner (3 min whole-body scan<0.1 mGy) per manufacturer protocol. A single technologist analyzed all DEXA measurements using Hologic Apex software (13.6.0.4:3) following the International Society for Clinical Densitometry guidelines. Visceral adipose tissue (VAT) was estimated from a 5-cm-wide region across the abdomen just above the iliac crest, coincident with the fourth lumbar vertebrae, to avoid interference from iliac crest bone pixels and matching the region commonly used to analyze VAT mass by CT scan (*Bredella et al., 2013*; *Kaul et al., 2012*; *Neeland et al., 2016*). The short version of the International Physical Activity Questionnaire (IPAQ) was used to assess the habitual physical activity levels of the participants. The IPAQ total score is expressed in metabolic equivalent (MET)-min/week (*Craig et al., 2003*).

## Dietary assessment

IDEO participants completed two dietary questionnaires, as previously described (*Alba et al., 2018*; *Oguri et al., 2020*), allowing for the assessment of usual total fiber intake and fiber from specific sources, as well as macronutrient, phytochemical, vitamin, and mineral uptake. The first instrument was an Automated Self-Administered 24 hr Dietary Assessment (ASA24) (*McClung et al., 2018*; *Park et al., 2018*; *Timon et al., 2016*), which queries intake over a 24-hr period. The 24 hr recalls and supplement data were manually entered in the ASA24 Dietary Assessment Tool (v. 2016), an electronic data collection and dietary analysis program. ASA24 employs research-based strategies to enhance dietary recall using a respondent-driven approach allowing initial recall to be self-defined. The second instrument was the National Cancer Institute's Diet History Questionnaire III (DHQIII) (*National Cancer Institute, 2020*; *Millen et al., 2006*). The DHQIII queries one's usual diet over the past month. Completing the DHQIII was associated with participant survey fatigue and completion rates were accordingly only 42% after one phone-based administration of the instrument, although they improved to 79% by the 2nd session and reached 100% within four sessions over a 5-month period. Due to the effort needed to achieve DHQIII completion, we modified our protocol to request completion of the simpler ASA24 at three separate times, at appointments where there were computers and personnel assistance for online completion, in addition to completion of the DHQIII questionnaire. By combining both instruments, we were able to reliably obtain complete dietary information on all participants.

## DNA extraction

Human stool samples were homogenized with bead beating for 5 min (Mini-Beadbeater-96, BioSpec) using beads of mixed size and material (Lysing Matrix E 2 ml Tube, MP Biomedicals) in the digestion solution and lysis buffer of a Wizard SV 96 Genomic DNA Kit (Promega). The samples were centrifuged for 10 min at 16,000×$g$ and the supernatant was transferred to the binding plate. The DNA was then purified according to the manufacturer's instructions. Mouse fecal pellets were homogenized with bead beating for 5 min (Mini-Beadbeater-96, BioSpec) using the ZR BashingBead lysis matrix containing 0.1 and 0.5 mm beads (ZR-96 BashingBead Lysis Rack, Zymo Research) and the lysis solution provided in the ZymoBIOMICS 96 MagBead DNA Kit (Zymo Research). The samples were centrifuged for 5 min at 3000×$g$ and the supernatant was transferred to 1 ml deep-well plates. The DNA was then purified using the ZymoBIOMICS 96 MagBead DNA Kit (Zymo Research) according to the manufacturer's instructions.

## 16S rRNA gene sequencing and analysis

For human samples, 16S rRNA gene amplification was carried out using GoLay-barcoded 515F/ 806R primers (*Caporaso et al., 2012*) targeting the V4 region of the 16S rRNA gene according to the methods of the Earth Microbiome Project (earthmicrobiome.org) (*Supplementary file 1B*). Briefly, 2 µl of DNA was combined with 25 µl of AmpliTaq Gold 360 Master Mix (Thermo Fisher Scientific), 5 µl of primers (2 µM each GoLay-barcoded 515/ 806R), and 18 µl H$_2$O. Amplification was as follows: 10 min 95°C, 30× (30 s 95°C, 30 s 50°C, 30 s 72°C), and 7 min 72°C. Amplicons were

quantified with PicoGreen (Quant-It dsDNA; Life Technologies) and pooled at equimolar concentrations. Aliquots of the pool were then column (MinElute PCR Purification Kit; Qiagen) and gel purified (QIAquick Gel Extraction Kit; Qiagen). Libraries were then quantified (KAPA Library Quantification Kit; Illumina) and sequenced with a 600 cycle MiSeq Reagent Kit (250×150; Illumina) with ~ 15% PhiX spike-in. For mouse samples, 16S rRNA gene amplification was carried out as per reference protocol and primers (*Gohl et al., 2016*). In brief, the V4 region of the 16S rRNA gene was amplified with 515F/806R primers containing common adaptor sequences, and then the Illumina flow cell adaptors and dual indices were added in a secondary amplification step (see *Supplementary file 1I* for index sequences). Amplicons were pooled and normalized using the SequalPrep Normalization Plate Kit (Invitrogen). Aliquots of the pool were then column (MinElute PCR Purification Kit, Qiagen) and gel purified (QIAquick Gel Extraction Kit, Qiagen). Libraries were then quantified and sequenced with a 600 cycle MiSeq Reagent Kit (270×270; Illumina) with ~ 15% PhiX spike-in.

Demultiplexed sequencing reads were processed using QIIME2 v2020.2 (*Bolyen et al., 2019*) with denoising by DADA2 (*Callahan et al., 2016*). Taxonomy was assigned using the DADA2 implementation of the RDP classifier (*Wang et al., 2007*) using the DADA2 formatted training sets for SILVA version 138 (benjjneb.github.io/dada2/assign.html). For ASV analyses, we utilized quality scores to set truncation and trim parameters. The reverse read of human 16S data suffered from low sequence quality and reduced the overall ASV counts, so we therefore analyzed only the forward reads, although a separate analysis using merged forward and reverse reads complemented the findings we report in this manuscript. For the manuscript, forward reads were truncated to 220 base pairs and underwent an additional 5 base pairs of trimming for 16S analysis of human stool. For gnotobiotic mice, forward and reverse reads were truncated to 200 and 150 base pairs, respectively. ASVs were filtered such that they were present in more than one sample with at least a total of 10 reads across all samples. Alpha diversity metrics were calculated on subsampled reads using Vegan (*Dixon, 2003*) and Picante (*Kembel et al., 2010*) R packages. The PhILR Euclidean distance was calculated by first carrying out the phylogenetic isometric log ratio transformation (philr, PhILR [*Silverman et al., 2017*]) followed by calculating the Euclidean distance (vegdist, Vegan [*Dixon, 2003*]). Principal coordinates analysis was carried out using the pcoa function of APE (*Paradis et al., 2004*). ADONIS calculations were carried out (adonis, Vegan) with 999 replications on each distance metric. The permutational space for the adonis calculation for the three combined gnotobiotic experiments was restricted by donor identifier to account for multiple recipient mice for a given donor and applied to *Figure 7—figure supplement 3* using setblocks to define permutations and specifying these blocks in the command. Centered $\log_2$-ratio (CLR) normalized abundances were calculated using the Make.CLR function in MicrobeR package (*Bisanz, 2017*) with count zero multiplicative replacement (zCompositions; *Martín-Fernández et al., 2014*). ALDEx2 (*Fernandes et al., 2013*) was used to analyze differential abundances of count data, using features that represented at least 0.05% of total sequencing reads. Corrections for multiple hypotheses using the Benjamini-Hochberg method (*Benjamini and Hochberg, 1995*) were performed where applicable. Where described, a false discovery rate (FDR) indicates the Benjamini-Hochberg adjusted p-value for an FDR (0.1 unless otherwise specified). Analysis of distance matrices and alpha diversity mirror prior analyses developed in the Turnbaugh lab and were adapted to the current manuscript (*Bisanz et al., 2019*). Calculations of associations between ASVs and ASA24 questionnaire data were completed by calculating a Spearman rank correlation and then adjusting the p-value for a Benjamini-Hochberg FDR using the cor_pmat function in the R package ggcorrplot (*Kassambara and Kassambara, 2019*) for all CLR transformed ASVs detected between ethnic groups. Shotgun data for each ethnicity was processed using Metaphlan2 and the species associations were calculated for relative abundance by ASA24 questionnaire data separate from the ASV data. The randomForest package (*Liaw and Wiener M, 2002*) was employed to generate random forest classifiers. Given the total number of samples (n=46), we generated 46 classifiers trained on a subset of 45 samples and used each classifier to predict the sample left out. AUCs are visualized utilizing the pROC (*Robin et al., 2011*) and ROCR (*Sing et al., 2005*) packages.

## Metagenomic sequencing and analysis

Whole-genome shotgun libraries were prepared using the Nextera XT DNA Library Prep Kit (Illumina). Paired ends of all libraries were sequenced on the NovaSeq 6000 platform in a single sequencing run (n=45 subjects; see *Supplementary file 1B* for relevant metadata and statistics). Illumina reads

underwent quality trimming and adaptor removal using fastp (*Chen et al., 2018*) and host read removal using BMTagger v1.1.0 (ftp.ncbi.nlm.nih.gov/pub/agarwala/bmtagger/) in the metaWRAP pipeline (github.com/bxlab/metaWRAP) (*Uritskiy et al., 2018*). Metagenomic samples were taxonomically profiled using MetaPhlan2 v2.7.7 (*Truong et al., 2015*) and functionally profiled using HUMAnN2 v0.11.2 (*Franzosa et al., 2018*), both with default parameters. Principal coordinates analysis on MetaPhlan2 species-level abundances was carried out using Bray Curtis distances and the pcoa function of APE (*Paradis et al., 2004*). Metaphlan2 abundance outputs were converted to counts and subsampled to even sample depth. Differences between groups were determined utilizing the Aldex2 package as described above. Tables of gene family abundances from HUMAnN2 were regrouped to KEGG orthologous groups using humann2_regroup_table. Functional pathways relating to SCFA production were manually curated from the pathway outputs from HUMANn2 and normalized by the estimated genome equivalents in each microbial community obtained from MicrobeCensus (*Nayfach and Pollard, 2015*).

## Quantification of bacterial load

Absolute 16S rRNA gene copy number was derived by adjustments for dilutions during DNA extraction and template normalization dividing by the total fecal mass used for DNA extraction in grams. Quantification of bacterial load was conducted using quantitative PCR (qPCR) given stool samples were frozen for the IDEO cohort as described above and bacterial lysis was achieved with a preparation including both bead beating and a detergent. Differences in 16S rRNA gene copy number between bacterial strains may have masked more subtle differences in colonization level. qPCR was performed on DNA extracted from the human stool samples. DNA templates were diluted 1:10 into a 96-well plate. Samples were aliquoted in a 384-well plate, and PCR primers and iTaq Universal Probes Supermix were added utilizing an Opentrons OT-2 instrument then analyzed on a Bio-Rad CFX384 thermocycler with an annealing temperature of 60°C . The following primers including a FAM labeled PCR probe was used for quantification: 891F , TGGAGCATGTGGTTTAATTCGA; 1003R , TGCGGGACTTAACCCAACA; 1002P , [6FAM]CACGAGCTGACGACARCCATGCA[BHQ1]. Absolute quantifications were determined against a standard curve of purified 8F /1542R amplified *Vibrio casei* DNA. Reactions identified as inappropriately amplified by the instrument were rejected, and the mean values were used for downstream analysis. Absolute 16S rRNA gene copy number was derived by adjustments for dilutions during DNA extraction and template normalization dividing by the total fecal mass used for DNA extraction in grams. Quantification of bacterial load was conducted using qPCR given stool samples were frozen for the IDEO cohort as described above and bacterial lysis was achieved with a preparation including both bead beating and a detergent.

## Nuclear magnetic resonance metabolomics

NMR spectroscopy was performed at 298K on a Bruker Avance III 600 MHz spectrometer configured with a 5 mm inverse cryogenic probe (Bruker Biospin, Germany) as previously described (*Cai et al., 2017*). Lean and obese EA and W individuals (n=20 total individuals, five in each group) were selected and matched based on body composition and metabolic parameters. Stool samples from these subjects were subjected to NMR-based metabolomics. 50 mg of human feces were extracted with 1 ml of phosphate buffer ($K_2HPO_4$/$NaH_2PO_4$, 0.1 M, pH 7.4, 50% v/v $D_2O$) containing 0.005% sodium 3-(trimethylsilyl) [2,2,3,3–2 H4] propionate (TSP-$d_4$) as a chemical shift reference (δ 0.00). Samples were freeze-thawed three times with liquid nitrogen and water bath for thorough extraction, then homogenized (6500 rpm, 1 cycle, 60 s) and centrifuged (11,180×$g$ , 4°C , 10 min). The supernatants were transferred to a new 2 ml tube. An additional 600 μl of phosphate-buffered saline was added to the pellets, followed by the same extraction procedure described above. Combined fecal extracts were centrifuged (11,180×$g$ , 4°C, 10 min), 600 μl of the supernatant was transferred to a 5 mm NMR tube (Norell, Morganton, NC) for NMR spectroscopy analysis. A standard one-dimensional NOESY pulse sequence noesypr1d (recycle delay-90°-t1-90°-tm-90°-acquisition) was used with a 90 pulse length of approximately 10 s (–9.6 dbW) and 64 transients are recorded into 32k data points with a spectral width of 9.6 kHz. NMR spectra were processed as previously described (*Cai et al., 2017*). First, spectra quality was improved with Topspin 3.0 (Bruker Biospin, Germany) for phase and baseline correction and chemical shift calibration. AMIX software (version: 3.9.14, Bruker Biospin, Germany)

was used for bucketing (bucket width 0.004 ppm), removal of interfering signal, and scaling (total intensity). Relative concentrations of identified metabolites were obtained by normalized peak area.

## Targeted gas chromatography mass spectrometry assays

Targeted analysis of SCFAs and BCAAs was performed with an Agilent 7890A gas chromatograph coupled with an Agilent 5975 mass spectrometer (Agilent Technologies, Santa Clara, CA) using a propyl esterification method as previously described (*Cai et al., 2017*). 50 mg of human fecal samples were pre-weighed, mixed with 1 ml of 0.005 M NaOH containing 10 μg/ml caproic acid-6,6,6-d3 (internal standard) and 1.0 mm diameter zirconia/silica beads (BioSpec, Bartlesville, OK). The mixture was thoroughly homogenized and centrifuged (13,200×$g$, 4°C, 20 min). 500 μl of supernatant was transferred to a 20 ml glass scintillation vial. 500 μl of 1-propanol/pyridine (v/v=3/2) solvent was added into the vial, followed by a slow adding of an aliquot of 100 μl of esterification reagent propyl chloroformate. After a brief vortex of the mixture for 1 min, samples were derivatized at 60°C for 1 hr. After derivatization, samples were extracted with hexane in a two-step procedure (300 μl + 200 μl) as described (*Zheng et al., 2013*). First, 300 μl of hexane was added to the sample, briefly vortexed and centrifuged (2000×$g$, 4°C, 5 min), and 300 μl of the upper layer was transferred to a glass autosampler vial. Second, an additional 200 μl of hexane was added to the sample, vortexed, centrifuged, and the 200 μl upper layer was transferred to the glass autosampler vial. A combination of 500 μl of extracts were obtained for GC-MS analysis. A calibration curve of each SCFA and BCAA was generated with series dilution of the standard for absolute quantitation of the biological concentration of SCFAs and BCAAs in human fecal samples.

## Targeted bile acid quantitation by UHPLC-MS/MS

Bile acid quantitation was performed with an ACQUITY ultra high pressure liquid chromatography (UHPLC) system using a Ethylene Bridged Hybrid C8 column (1.7 μm, 100 mm×2.1 mm) coupled with a Xevo TQ-S mass spectrometer equipped with an electrospray ionization source operating in negative mode (All Waters, Milford, MA) as previously described (*Sarafian et al., 2015*). Selected ion monitoring for non-conjugated bile acids and multiple reaction monitoring for conjugated bile acids was used. 50 mg of human fecal sample was pre-weighed, mixed with 1 ml of pre-cooled methanol containing 0.5 μM of stable-isotope-labeled bile acids (internal standards), and 1.0 mm diameter zirconia/silica beads (BioSpec, Bartlesville, OK), followed by thorough homogenization and centrifugation. Supernatant was transferred to an autosampler vial for analysis. 100 μl of serum was extracted by adding 200 μl pre-cooled methanol containing 0.5 μM deuterated bile acids as internal standards. Following centrifugation, the supernatant of the extract was transferred to an autosampler vial for quantitation. Calibration curves of individual bile acids were drafted with bile acid standards for quantitation of the biological abundance of bile acids.

## Gnotobiotic mouse experiments

All mouse experiments were approved by the UCSF Institutional Animal Care and Use Committee and performed accordingly. Germ-free mice were maintained within the UCSF Gnotobiotic Core Facility and fed ad libitum autoclaved standard chow diet (Lab Diet 5021). Germ-free adult male C57BL/6J mice between 6 and 10 weeks of age were used for all the experiments described in this paper. 10 lean subjects in our IDEO cohort were selected as donors for the microbiota transplantation experiments, including 5 EA and 5 W donors. The selected donors for gnotobiotic experiments were matched for phenotypic data to the degree possible (*Supplementary file 1H*). Stool samples to be used for transplantation were resuspended in 10 volumes (by weight) of brain heart infusion media in an anaerobic Coy chamber. Each diluted sample was vortexed for 1 min and left to settle for 5 min, and a single 200 μl aliquot of the clarified supernatant was administered by oral gavage into each germ-free mouse recipient. In experiments LFPP1 and LFPP2, microbiome transplantations were performed for two donors per experiment (1 W, 1 EA) with gnotobiotic mice housed in sterile isolators (CBC flexible, softwall isolator) and maintained on ad libitum standard chow also known as low-fat, high-plant-polysaccharide (LFPP) diet. In LFPP1, six germ-free mice per colonization group received an aliquot of stool from a donor of either ethnicity and body composition (measured using EchoMRI) were recorded on the day of colonization and at 6 weeks post-transplantation (per group n=6 recipient mice, 1 isolator, 2 cages). In LFPP2, we shortened the colonization time to 3 weeks and

used two new donor samples. For the third experiment (HFHS experiment), mice were weaned onto an irradiated HFHS diet (TD.88137, Envigo) for 4 weeks prior to colonization and housed in pairs in Tecniplast IsoCages. The same four donors from LFPP1 and LFPP2 were included in the HFHS experiment, in addition to six new donors (per donor n=2 recipient mice, 1 IsoCage). Body weight and body composition were recorded on the day of colonization and again at 3 weeks post-transplantation. Mice were maintained on the HFHS diet throughout the experiment. All samples were sequenced in a single pool (*Supplementary file 1I*). For comparisons between donors and recipient mice, donors and recipient mice were subsampled to even sequencing depth and paired between donor and recipient mice (range: 18,544–78,361 sequencing reads/sample).

## Glucose tolerance tests

Food was removed from mice 10 hr (LFPP1 experiment) or 4 hr (HFHS experiment) prior to assessment of glucose tolerance. Mice received i.p. injections of D-glucose (2 mg/kg), followed by the repeated collection of blood by tail nick and determination of glucose levels by handheld glucometer (Abbott Diabetes Care) over a 2-hr period.

## Geographic analyses

Map tiles and distance data were obtained using GGMap (*Kahle and Wickham, 2013*), OpenStreet Maps (*Fellows and Stotz, 2016*), and the Imap R (*Wallace, 2012*) packages. GGMap was employed using a Google Cloud API key and the final map tiles were obtained in July 2020 (*Kahle and Wickham, 2013*). Spearman ranked correlation coefficients (*rho*) were calculated as embedded in the ggpubr (*Kassambara, 2018*) R package. 2018 US Census data for EA and W subjects was obtained (B02001 table for race, data.census.gov) for the ZIP codes available in our study and using the leaflet (*Cheng et al., 2018*) package. The census data used is included as part of *Supplementary file 1B* to aid in reproduction. Each census region is plotted as a percentage of W individuals over a denominator of W and EA subjects. The leaflet package utilized ZIP Code Tabulation Areas (ZCTAs) from the 2010 census. We extracted all ZCTAs starting with 9, and the resulting 29 ZIP codes that overlap with IDEO subjects were analyzed (*Supplementary file 1B*). Two ZCTAs (95687 and 95401) were primarily W when comparing W and EA subjects. There were two W subjects recruited from these ZTCAs. These ZIP codes are cutoff based on the zoom magnification for that figure and as a result ZTCAs for 27 individuals are plotted. Distance to a central point in SF was calculated. The point of reference was latitude=37.7585102, longitude=−122.4539916.

## Dietary questionnaire correlation analysis

DHQIII and ASA24 data were analyzed using a Euclidean distance matrix. These transformations were completed using the cluster package (*Maechler et al., 2021*). Subsequent analysis was completed using the vegan package (*Dixon, 2003*; *Oksanen et al., 2013*). Procrustes transformations were performed using 16S-seq data from human subjects, which was then subjected to a PhILR transformation. The resulting matrix was rotated against the distance matrix for ASA24 or DHQIII questionnaire data using the procrustes command in the vegan R package using 999 permutations. Mantel statistics were calculated utilizing the mantel command of the vegan package.

## R packages used in this study

Picante (*Kembel et al., 2010*), PhILR (*Silverman et al., 2017*), MicrobeR (*Bisanz, 2017*), ALDEx2 (*Fernandes et al., 2013*), ggcorrplot (*Kassambara and Kassambara, 2019*), randomForest (*Liaw and Wiener M, 2002*), GGMap (*Kahle and Wickham, 2013*), OpenStreetMap (*Fellows and Stotz, 2016*), IMap (*Wallace, 2012*), ggpubr (*Kassambara, 2018*), leaflet (*Cheng et al., 2018*), cluster (*Maechler et al., 2021*), readxl (*Wickham and Bryan, 2017*), Rtsne (*Krijthe, 2015*), vegan (*Dixon, 2003*; *Oksanen et al., 2013*), ape (*Paradis and Schliep, 2019*), tigris (*Walker, 2018*), lmerTest (*Kuznetsova et al., 2017*), qiime2R (*Bisanz, 2018*), gghighlight (*Yutani, 2018*), Phyloseq (*McMurdie et al., 2013*), Janitor (*Firke, 2018*), table1 (*Rich, 2020*), and ggplot2 (*Wickham, 2016*).

## Statistical analyses

Statistical analysis of the human data was performed using the table1 package in R (STATCorp LLC, College Station, TX). Human data were presented as mean ± SD. Unpaired independent Student's

t-tests were used to compare differences between the two groups in the case of continuous data and in the case of categorical data the $\chi^2$ test was utilized for *Supplementary file 1A*. For a given lean or obese categories between ethnicity tests were adjusted for a Benjamini-Hochberg FDR utilizing the command p.adjust in R, which is indicated as an adjusted p-value in the tables and none were significant as described in the table legend. In *Supplementary file 1G,H*, no values met an adjusted p-value cutoff of <0.1. In *Supplementary file 1A* and p-values indicated by numbers were pooled together for adjustments and those represented by symbols were separately pooled together for adjustment. All microbiome-related analyses were carried out in R version 3.5.3 or 4.0.2. Where indicated, Wilcoxon rank-sum tests were calculated. A Benjamini-Hochberg adjusted p-value (FDR) of 0.1 was used as the cutoff for statistical significance unless stated otherwise. Statistical analysis of glucose tolerance tests was carried out using linear mixed-effects models with the lmerTest (*Kuznetsova et al., 2017*) R package and mouse as random effect. Graphical representation was carried out using ggplot2. Boxplots indicate the interquartile range (25th to 75th percentiles), with the center line indicating the median and whiskers representing 1.5× the interquartile range.

## Availability of data and materials

All 16S-seq and metagenomic sequencing data generated in the preparation of this manuscript have been deposited in NCBI's Sequence Read Archive under accession number PRJNA665061. Metabolomics results and metadata are available within this manuscript (*Supplementary file 1*). Code for our manuscript and a more comprehensive metadata table is available on GitHub (https://github.com/turnbaughlab/2021_IDEO, *Upadhyay and Turnbaugh, 2021*; copy archived at swh:1:rev:07f9ee797d57620e10734bef4d893bf51662559c).

## Acknowledgements

The authors thank Jessie Turnbaugh and the other UCSF Gnotobiotics Core Facility staff and members of the Koliwad lab for help with the gnotobiotic mouse experiments. The authors thank Dr. Philip B Smith from the Penn State Metabolomics Facility. The authors also thank the CZ Biohub Sequencing Platform for sequencing support, as well as all the subjects who participated in this study.

## Additional information

### Competing interests

Peter J Turnbaugh: is on the scientific advisory board for Kaleido, Pendulum, Seres, and SNIPRbiome; there is no direct overlap between the current study and these consulting duties. Reviewing editor, eLife. The other authors declare that no competing interests exist.

### Funding

| Funder | Grant reference number | Author |
| --- | --- | --- |
| National Institute of Diabetes and Digestive and Kidney Diseases | R01DK114034 | Peter J Turnbaugh |
| National Institute of Diabetes and Digestive and Kidney Diseases | R01DK11230401 | Suneil K Koliwad |
| National Institutes of Health | R01HL122593 | Peter Turnbaugh |
| National Institutes of Health | R01AR074500 | Peter Turnbaugh |
| National Institutes of Health | R01DK11230403S1 | Suneil K Koliwad |
| National Institutes of Health | P30DK098722 | Suneil K Koliwad |

| Funder | Grant reference number | Author |
|---|---|---|
| American Diabetes Association | 1-18-PMF-003 | Diana L Alba |
| National Institutes of Health | T32HL007185 | Vaibhav Upadhyay |

The funders had no role in study design, data collection and interpretation, or the decision to submit the work for publication.

## Author contributions
Qi Yan Ang, Conceptualization, Data curation, Formal analysis, Investigation, Methodology, Visualization, Writing – original draft, Writing – review and editing; Diana L Alba, Conceptualization, Data curation, Formal analysis, Investigation, Methodology, Writing – original draft, Writing – review and editing; Vaibhav Upadhyay, Data curation, Formal analysis, Investigation, Methodology, Validation, Visualization, Writing – original draft, Writing – review and editing; Jordan E Bisanz, Cecilia Noecker, Methodology, Software; Jingwei Cai, Data curation, Methodology; Ho Lim Lee, Eliseo Barajas, Data curation; Grace Wei, Data curation, Resources; Andrew D Patterson, Methodology, Supervision; Suneil K Koliwad, Conceptualization, Funding acquisition, Project administration, Resources, Supervision, Writing – review and editing; Peter J Turnbaugh, Conceptualization, Funding acquisition, Project administration, Resources, Supervision, Writing – original draft, Writing – review and editing

## Author ORCIDs
Qi Yan Ang (iD) http://orcid.org/0000-0002-9788-1283
Diana L Alba (iD) http://orcid.org/0000-0002-1398-9861
Vaibhav Upadhyay (iD) http://orcid.org/0000-0002-7652-1043
Jordan E Bisanz (iD) http://orcid.org/0000-0002-8649-1706
Peter J Turnbaugh (iD) http://orcid.org/0000-0002-0888-2875

## Ethics
Human stool samples were collected as part of a multi-ethnic clinical cohort study termed Inflammation, Diabetes, Ethnicity and Obesity (ClinicalTrials.gov identifier NCT03022682), consisting of 25- to 65-year-old men and women residing in Northern California and recruited from medical and surgical clinics at UCSF and the Zuckerberg San Francisco General Hospital, or through local public advertisements. The host phenotypic data from this cohort have been described in detail (*Alba et al., 2018*; *Oguri et al., 2020*). Informed consent was obtained from all subjects participating in the study, which was approved by the UCSF Institutional Review Board (IRB #14–14248).

Protocols for all experiments involving mice were approved by the University of California, San Francisco Institutional Animal Care and Use Committee, and performed accordingly (UCSF IACUC numbers AN183950 and AN184143).

## Decision letter and Author response
Decision letter https://doi.org/10.7554/eLife.70349.sa1
Author response https://doi.org/10.7554/eLife.70349.sa2

# Additional files

## Supplementary files
• Supplementary file 1. This file contains 10 supplementary tables that include detailed metadata, metabolomics data, and data visualized in the main text and supplement figures for the manuscript.
• Transparent reporting form

## Data availability
All 16S-seq and metagenomic sequencing data generated in the preparation of this manuscript have been deposited in NCBI's Sequence Read Archive under accession number PRJNA665061. Metabolomics results and metadata are available within this manuscript (Supplementary File 1). Code for our manuscript and a more comprehensive metadata table is available on GitHub (https://github.com/

turnbaughlab/2021_IDEO; copy archived at https://archive.softwareheritage.org/swh:1:rev:07f9ee79 7d57620e10734bef4d893bf51662559c).

The following dataset was generated:

| Author(s) | Year | Dataset title | Dataset URL | Database and Identifier |
|---|---|---|---|---|
| Ang QY, Alba DL, Upadhyay V, Koliwad SK, Turnbaugh PJ | 2020 | IDEO Microbiome Study | https://www.ncbi.nlm. nih.gov/bioproject/? term=PRJNA665061 | NCBI BioProject, PRJNA665061 |

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
