## [Decision Letter]

**Acceptance summary:**

Thank you very much for an extremely careful revision and attending to the points in the critiques. I have greatly enjoyed re-reading the paper which I see as a very useful assessment of microbiome and stool metabolome differences that incorporate stratification according to ethnicity, obesity, medication differences and geographic location. The fact that you show that differences can be modeled after transplantation into germ-free mice can be an entry point into subsequent mechanistic determination. The differences between the East Asians and White subjects living in the same country that show different sensitivity of clinical phenotypes (BMI, body fat) according to microbial richness also seems an entry point into understanding the impact of microbiota, diet, other environmental factors and host genetics on disease susceptibility.

**Decision letter after peer review:**

Thank you for submitting your article "The East Asian Gut Microbiome is Distinct From Colocalized White Subjects and Connected to Metabolic Health" for consideration by *eLife*. Your article has been reviewed by 3 peer reviewers, one of whom is a member of our Board of Reviewing Editors, and the evaluation has been overseen by Gisela Storz as the Senior Editor. The following individual involved in review of your submission has agreed to reveal their identity: Bahtiyar Yilmaz (Reviewer #3).

Essential revisions:

1) In Table 1 the significance of differences between the East Asian and white ethnicities is given as adjusted P values. This potentially obscures a significant difference between the groups in single metrics that are then used to stratify later analysis. For example it appears that BMI and weight may be significantly different in the obese East Asian and white subjects, and that the obese East Asians are taking less exercise.

2) Could you please provide more information on the setting of recruitment and justify the limitations of bias in the dataset? It also sounds as if there is a far more detailed clinical metadataset available for these subjects (smoking, medications, parameters summarized in Figure S4). It would help other investigators to have this data available on an (anonymized) per subject basis. The conclusions include (correct) statements about the importance of the IDEO cohort, but the metadata is important for cross-comparison between studies.

3) In Figure 1, you showed that richness and diversity are higher in W than EA. However, all the major phyla are higher in EA population. Even though you showed the statistical differences between tested group, please add the taxa bar plot to show the overall picture of the groups, preferably at phylum and genus level.

4) It is difficult to appreciate the timing of the output of the transfer experiments in 7A and E. Depending on the direction of the trajectory in 7A (do the samples generally separate or converge?), what happens if you down-sample to remove the samples in the bottom left of the ordination plot?

5) The interpretation in the last sentence of the discussion that the transfer experiments support the redundancy of host genetics in maintaining the differences depends on the absence of stochastic effects during the transfer (i.e. that replicate controls from the same subject map together by comparison). It is suggested that the interpretation could be offered with this caveat.

---

## [Author Response]

Essential revisions:1) In Table 1 the significance of differences between the East Asian and white ethnicities is given as adjusted P values. This potentially obscures a significant difference between the groups in single metrics that are then used to stratify later analysis. For example it appears that BMI and weight may be significantly different in the obese East Asian and white subjects, and that the obese East Asians are taking less exercise.

We have modified our manuscript as suggested: Table S1 now displays unadjusted *p*-values, while also referring to our multiple testing correction in the footnote. As expected, switching to unadjusted (nominal) *p*-values revealed more variables that are significantly different between groups; however, these differences were mainly restricted to *obese* EA and W individuals. This observation fits with the prior literature wherein differences in body composition and health related BMI outcomes are well-established and provided the rationale for our study (discussed in our *Introduction*).

In contrast, we only identified a single variable (triglyceride levels) that was significantly different between lean EA and W individuals. Thus, the central finding of our manuscript revealing variations in the gut microbiotas of *lean* EA and W subjects is only confounded by a single difference in triglyceride levels. Furthermore, the *p*-value was borderline (*p=*0.02) and did not survive multiple hypothesis testing.

We have modified our main text to reflect this new information:

“While several factors linked to body composition were different between obese EA and W subjects using a nominal p-value, only triglyceride levels were significantly different between lean EA and W subjects and this trend did not survive multiple testing correction (Table S1).”

2) Could you please provide more information on the setting of recruitment and justify the limitations of bias in the dataset? It also sounds as if there is a far more detailed clinical metadataset available for these subjects (smoking, medications, parameters summarized in Figure S4). It would help other investigators to have this data available on an (anonymized) per subject basis. The conclusions include (correct) statements about the importance of the IDEO cohort, but the metadata is important for cross-comparison between studies.

As requested, we have expanded our methods section to provide a more detailed discussion of recruitment and justify limitations of bias in our dataset (copied below). We have also provided more comprehensive metadata linked to the participants we recruited into this study from IDEO (IDEO_Extended_Data.csv). This includes data on smoking, medications, and all of the parameters summarized in Figure S4 as the reviewers requested. A subset of this data was included in Table S2 in our original submission. We have preserved all of the data in Table S2 in its original form and supplemented all of the additional requested data, including 404 non-redundant metadata variables, in the IDEO_Extended_Data.csv file, which is posted to our laboratory’s github repository for this manuscript.

“Human subjects

The IDEO cohort was established to explore the pathogenesis of obesity and metabolic diseases in highly vulnerable segments of the population. […] Samples were aliquoted and stored at -80°C.”

3) In Figure 1, you showed that richness and diversity are higher in W than EA. However, all the major phyla are higher in EA population. Even though you showed the statistical differences between tested group, please add the taxa bar plot to show the overall picture of the groups, preferably at phylum and genus level.

We have updated Figure 1C. We opted to present the phylum data as box plots to be consistent with the CLR-transformed values of taxonomic abundances that we used throughout the figures. CLR values (which are on a log2 scale) cannot be stacked in the same way as percent relative abundances. In our opinion, stacked bar graphs are also difficult to interpret due to the small size of each bar, the high abundance of the top 2 phyla relative to the others, and the lack of any indication of variance within a group.

We think that the expanded Figure 1C makes the overall picture at the phylum level quite clear. In our previous submission, we highlighted 6 bacterial phyla that are significantly different between ethnicity, of which only one, *Verrucomicrobiota*, was significantly enriched in W subjects. In this updated figure, we show the other bacterial phyla, including *Desulfobacterota* and *Cyanobacteria*, which trends to be higher in W subjects though not individually significant. The abundances of the non-significant phyla, collectively, are higher in W compared to EA subjects (*p* = 0.012, Wilcoxon rank-sum test). We discuss these results in the main text as follows:

“Phylogenetic analyses of all ASVs revealed marked variations in the direction of change across different phyla between EA and W subjects (Figure S1A), indicating that the phylum level trends (Figure 1C) resulted from the integration of subtle shifts across multiple component members (Figure 1D-F).”

As requested, we have also included an additional panel, Figure 1D, showing stacked bar plots of average relative abundances at the genus level. Consistent with the observation that W subjects have higher microbial diversity and lower relative abundances of the major phyla compared to EA subjects, the bar plots show that W subjects have higher relative abundances of the lower abundance taxa (merged into a single category as “Remainder” in the bar plots).

4) It is difficult to appreciate the timing of the output of the transfer experiments in 7A and E. Depending on the direction of the trajectory in 7A (do the samples generally separate or converge?), what happens if you down-sample to remove the samples in the bottom left of the ordination plot?

We apologize for the confusion. The samples shown in Figure 7A represent gnotobiotic mice that are connected by the same donor, not longitudinal samples from the same mice. Samples were collected at the end of each experiment (6 weeks in the case of LFPP1, 3 weeks in the case of LFPP2, and 4 weeks in the case of the HFHS experiment as described in the methods section of our manuscript – only *one* time point per experiment). Because these samples do not represent different time points from the same experiment, all of them are required for comparisons between ethnicities.

Figure 7A combines results from two independent experiments where microbial communities of the recipient mice largely cluster by sample donor and donor ethnicity. In a prior version of this manuscript, we had separated these two LFPP experiments; however, we were encouraged by our previous reviewers to focus on trends that are consistent across the two replicate experiments, providing the rationale for the current figure.

For your information, we have re-separated these two experiments into two separate LFPP experiments (see Author response image 1) . The samples described in the reviewer question are represented by the EA recipients in the new LFPP1 plot in (Author response image 1) (the blue dots). We hope this clarifies our experimental design. Given that these are not multiple timepoints and because comparisons between all recipient mice between both ethnicities is appropriate and was recommended during our initial round of review, we have opted to include all samples and experiments in the main text.

**Author response image 1. sa2fig1:** Principal coordinate analysis of PhILR Euclidean distances of stool from germ-free recipient mice transplanted with stool microbial communities from lean EA or W donors and fed a LFPP diet in two independent experiments (LFPP1 and LFPP2, n=12 recipient mice per group in each experiment). Significance was assessed by ADONIS. Each data point represents a single mouse. Germ-free mice receiving the same donor sample are connected by a dashed line. Experimental designs are shown in Figure S7.

5) The interpretation in the last sentence of the discussion that the transfer experiments support the redundancy of host genetics in maintaining the differences depends on the absence of stochastic effects during the transfer (i.e. that replicate controls from the same subject map together by comparison). It is suggested that the interpretation could be offered with this caveat.

We think that this comment refers to the final sentence in the first paragraph of our original discussion, which read:

“Our experiments using inbred germ-free mice support the stability of ethnicity-associated differences in the gut microbiota on both the LFPP and HFHS diets, while also demonstrating that variations in host genetics are not necessary to maintain these signatures, at least over short timescales.”

We considered potential stochastic effects during our experimental design and data analysis. Although we conducted three different experiments with varied conditions and detected the impact of ethnicity on gut microbial community composition, we cannot fully exclude that stochastic effects linked to gnotobiotic transplantation impacted our results. We have modified the last sentence of the first paragraph of the discussion to include this caveat and related points:

“Our experiments using inbred germ-free mice support the stability of ethnicity-associated differences in the gut microbiota on both the LFPP and HFHS diets, while also demonstrating that variations in host genetics are not necessary to maintain these signatures, at least over short timescales. Even though we conducted multiple experiments and recipient mice from the same donor generally mapped together, differences between the human donor and recipient mouse microbiotas inherent to gnotobiotic transplantation warrant further investigation…”

References:

Alba DL, Farooq JA, Lin MYC, Schafer AL, Shepherd J, Koliwad SK. 2018. Subcutaneous Fat Fibrosis Links Obesity to Insulin Resistance in Chinese Americans. J Clin Endocrinol Metab 103:3194–3204.

American Diabetes Association. 2019. 2. Classification and Diagnosis of Diabetes: Standards of Medical Care in Diabetes—2019. Diabetes Care 42:S13–S28.

Arumugam M, Raes J, Pelletier E, Le Paslier D, Yamada T, Mende DR, Fernandes GR, Tap J, Bruls T, Batto J-M, Bertalan M, Borruel N, Casellas F, Fernandez L, Gautier L, Hansen T, Hattori M, Hayashi T, Kleerebezem M, Kurokawa K, Leclerc M, Levenez F, Manichanh C, Nielsen HB, Nielsen T, Pons N, Poulain J, Qin J, Sicheritz-Ponten T, Tims S, Torrents D, Ugarte E, Zoetendal EG, Wang J, Guarner F, Pedersen O, de Vos WM, Brunak S, Doré J, MetaHIT Consortium, Antolín M, Artiguenave F, Blottiere HM, Almeida M, Brechot C, Cara C, Chervaux C, Cultrone A, Delorme C, Denariaz G, Dervyn R, Foerstner KU, Friss C, van de Guchte M, Guedon E, Haimet F, Huber W, van Hylckama-Vlieg J, Jamet A, Juste C, Kaci G, Knol J, Lakhdari O, Layec S, Le Roux K, Maguin E, Mérieux A, Melo Minardi R, M’rini C, Muller J, Oozeer R, Parkhill J, Renault P, Rescigno M, Sanchez N, Sunagawa S, Torrejon A, Turner K, Vandemeulebrouck G, Varela E, Winogradsky Y, Zeller G, Weissenbach J, Ehrlich SD, Bork P. 2011. Enterotypes of the human gut microbiome. Nature 473:174–180.

Basolo A, Hohenadel M, Ang QY, Piaggi P, Heinitz S, Walter M, Walter P, Parrington S, Trinidad DD, von Schwartzenberg RJ, Turnbaugh PJ, Krak_off_ J. 2020. Effects of underfeeding and oral vancomycin on gut microbiome and nutrient absorption in humans. Nat Med 26:589–598.

Berding K, Donovan SM. 2018. Diet Can Impact Microbiota Composition in Children With Autism Spectrum Disorder. Front Neurosci 12:515.

Bisanz JE, Upadhyay V, Turnbaugh JA, Ly K, Turnbaugh PJ. 2019. Meta-Analysis Reveals Reproducible Gut Microbiome Alterations in Response to a High-Fat Diet. Cell Host Microbe 26:265–272.e4.

Brown K, Godovannyi A, Ma C, Zhang Y, Ahmadi-Vand Z, Dai C, Gorzelak MA, Chan Y, Chan JM, Lochner A, Dutz JP, Vallance BA, Gibson DL. 2016. Prolonged antibiotic treatment induces a diabetogenic intestinal microbiome that accelerates diabetes in NOD mice. ISME J 10:321–332.

Cai J, Zhang L, Jones RA, Correll JB, Hatzakis E, Smith PB, Gonzalez FJ, Patterson AD. 2016. Antioxidant Drug Tempol Promotes Functional Metabolic Changes in the Gut Microbiota. J Proteome Res 15:563–571.

Depommier C, Everard A, Druart C, Plovier H, Van Hul M, Vieira-Silva S, Falony G, Raes J, Maiter D, Delzenne NM, de Barsy M, Loumaye A, Hermans MP, Thissen J-P, de Vos WM, Cani PD. 2019. Supplementation with Akkermansia muciniphila in overweight and obese human volunteers: a proof-of-concept exploratory study. Nat Med 25:1096–1103.

Ding T, Schloss PD. 2014. Dynamics and associations of microbial community types across the human body. Nature 509:357–360.

Durazzi F, Sala C, Castellani G, Manfreda G, Remondini D, De Cesare A. 2021. Comparison between 16S rRNA and shotgun sequencing data for the taxonomic characterization of the gut microbiota. Sci Rep 11:3030.

Falony G, Joossens M, Vieira-Silva S, Wang J, Darzi Y, Faust K, Kurilshikov A, Bonder MJ, Valles-Colomer M, Vandeputte D, Tito RY, Chaffron S, Rymenans L, Verspecht C, De Sutter L, Lima-Mendez G, D’hoe K, Jonckheere K, Homola D, Garcia R, Tigchelaar EF, Eeckhaudt L, Fu J, Henckaerts L, Zhernakova A, Wijmenga C, Raes J. 2016. Population-level analysis of gut microbiome variation. Science 352:560–564.

Forslund K, Hildebrand F, Nielsen T, Falony G, Le Chatelier E, Sunagawa S, Prifti E, Vieira-Silva S, Gudmundsdottir V, Pedersen HK, Arumugam M, Kristiansen K, Voigt AY, Vestergaard H, Hercog R, Costea PI, Kultima JR, Li J, Jørgensen T, Levenez F, Dore J, MetaHIT consortium, Nielsen HB, Brunak S, Raes J, Hansen T, Wang J, Ehrlich SD, Bork P, Pedersen O. 2015. Disentangling type 2 diabetes and metformin treatment signatures in the human gut microbiota. Nature 528:262–266.

Gaulke CA, Sharpton TJ. 2018. The influence of ethnicity and geography on human gut microbiome composition. Nat Med.

Ghosh TS, Das M, Jeffery IB, O’Toole PW. 2020. Adjusting for age improves identification of gut microbiome alterations in multiple diseases. *eLife* 9. doi:10.7554/*eLife.*50240

Gu D, He J, Duan X, Reynolds K, Wu X, Chen J, Huang G, Chen C-S, Whelton PK. 2006. Body weight and mortality among men and women in China. JAMA 295:776–783.

He Y, Wu W, Zheng H-M, Li P, McDonald D, Sheng H-F, Chen M-X, Chen Z-H, Ji G-Y, Zheng Z-D-X, Mujagond P, Chen X-J, Rong Z-H, Chen P, Lyu L-Y, Wang X, Wu C-B, Yu N, Xu Y-J, Yin J, Raes J, Knight R, Ma W-J, Zhou H-W. 2018. Regional variation limits applications of healthy gut microbiome reference ranges and disease models. Nat Med 24:1532–1535.

Hsu WC, Araneta MRG, Kanaya AM, Chiang JL, Fujimoto W. 2015. BMI cut points to identify at-risk Asian Americans for type 2 diabetes screening. Diabetes Care 38:150–158.

Jih J, Mukherjea A, Vittinghoff E, Nguyen TT, Tsoh JY, Fukuoka Y, Bender MS, Tseng W, Kanaya AM. 2014. Using appropriate body mass index cut points for overweight and obesity among Asian Americans. Prev Med 65:1–6.

Johnson AJ, Vangay P, Al-Ghalith GA, Hillmann BM, Ward TL, Shields-Cutler RR, Kim AD, Shmagel AK, Syed AN, Personalized Microbiome Class Students, Walter J, Menon R, Koecher K, Knights D. 2019. Daily Sampling Reveals Personalized Diet-Microbiome Associations in Humans. Cell Host Microbe 25:789–802.e5.

Johnson EL, Heaver SL, Walters WA, Ley RE. 2017. Microbiome and metabolic disease: revisiting the bacterial phylum Bacteroidetes. J Mol Med 95:1–8.

Kassambara A, Kassambara MA. 2019. Package “ggcorrplot.” R package version 0 1 3.

Kim KN, Yao Y, Ju SY. 2019. Short Chain Fatty Acids and Fecal Microbiota Abundance in Humans with Obesity: A Systematic Review and Meta-Analysis. Nutrients 11. doi:10.3390/nu11102512

Liu R, Hong J, Xu X, Feng Q, Zhang D, Gu Y, Shi J, Zhao S, Liu W, Wang X, Xia H, Liu Z, Cui B, Liang P, Xi L, Jin J, Ying X, Wang X, Zhao X, Li W, Jia H, Lan Z, Li F, Wang R, Sun Y, Yang M, Shen Y, Jie Z, Li J, Chen X, Zhong H, Xie H, Zhang Y, Gu W, Deng X, Shen B, Xu X, Yang H, Xu G, Bi Y, Lai S, Wang J, Qi L, Madsen L, Wang J, Ning G, Kristiansen K, Wang W. 2017. Gut microbiome and serum metabolome alterations in obesity and after weight-loss intervention. Nat Med 23:859–868.

Lu Y, Fan C, Li P, Lu Y, Chang X, Qi K. 2016. Short Chain Fatty Acids Prevent High-fat-diet-induced Obesity in Mice by Regulating G Protein-coupled Receptors and Gut Microbiota. Sci Rep 6:37589.

Lynch CJ, Adams SH. 2014. Branched-chain amino acids in metabolic signalling and insulin resistance. Nat Rev Endocrinol 10:723–736.

Müller M, Hernández MAG, Goossens GH, Reijnders D, Holst JJ, Jocken JWE, van Eijk H, Canfora EE, Blaak EE. 2019. Circulating but not faecal short-chain fatty acids are related to insulin sensitivity, lipolysis and GLP-1 concentrations in humans. Sci Rep 9:12515.

Nayak RR, Alexander M, Deshpande I, Stapleton-Gray K, Rimal B, Patterson AD, Ubeda C, Scher JU, Turnbaugh PJ. 2021. Methotrexate impacts conserved pathways in diverse human gut bacteria leading to decreased host immune activation. Cell Host Microbe. doi:10.1016/j.chom.2020.12.008

Oguri Y, Shinoda K, Kim H, Alba DL, Bolus WR, Wang Q, Brown Z, Pradhan RN, Tajima K, Yoneshiro T, Ikeda K, Chen Y, Cheang RT, Tsujino K, Kim CR, Greiner VJ, Datta R, Yang CD, Atabai K, McManus MT, Koliwad SK, Spiegelman BM, Kajimura S. 2020. CD81 Controls Beige Fat Progenitor Cell Growth and Energy Balance via FAK Signaling. Cell 182:563–577.e20.

Perry RJ, Peng L, Barry NA, Cline GW, Zhang D, Cardone RL, Petersen KF, Kibbey RG, Goodman AL, Shulman GI. 2016. Acetate mediates a microbiome–brain–β-cell axis to promote metabolic syndrome. Nature 534:213–217.

Plovier H, Everard A, Druart C, Depommier C, Van Hul M, Geurts L, Chilloux J, Ottman N, Duparc T, Lichtenstein L, Myridakis A, Delzenne NM, Klievink J, Bhattacharjee A, van der Ark KCH, Aalvink S, Martinez LO, Dumas M-E, Maiter D, Loumaye A, Hermans MP, Thissen J-P, Belzer C, de Vos WM, Cani PD. 2017. A purified membrane protein from Akkermansia muciniphila or the pasteurized bacterium improves metabolism in obese and diabetic mice. Nat Med 23:107–113.

Qin J, Li Y, Cai Z, Li S, Zhu J, Zhang F, Liang S, Zhang W, Guan Y, Shen D, Peng Y, Zhang D, Jie Z, Wu W, Qin Y, Xue W, Li J, Han L, Lu D, Wu P, Dai Y, Sun X, Li Z, Tang A, Zhong S, Li X, Chen W, Xu R, Wang M, Feng Q, Gong M, Yu J, Zhang Y, Zhang M, Hansen T, Sanchez G, Raes J, Falony G, Okuda S, Almeida M, LeChatelier E, Renault P, Pons N, Batto J-M, Zhang Z, Chen H, Yang R, Zheng W, Li S, Yang H, Wang J, Ehrlich SD, Nielsen R, Pedersen O, Kristiansen K, Wang J. 2012. A metagenome-wide association study of gut microbiota in type 2 diabetes. Nature 490:55–60.

Quast C, Pruesse E, Yilmaz P, Gerken J, Schweer T, Yarza P, Peplies J, Glöckner FO. 2013. The SILVA ribosomal RNA gene database project: improved data processing and web-based tools. Nucleic Acids Res 41:D590–6.

Ranjan R, Rani A, Metwally A, McGee HS, Perkins DL. 2016. Analysis of the microbiome: Advantages of whole genome shotgun versus 16S amplicon sequencing. Biochem Biophys Res Commun 469:967–977.

Schnorr SL, Candela M, Rampelli S, Centanni M, Consolandi C, Basaglia G, Turroni S, Biagi E, Peano C, Severgnini M, Fiori J, Gotti R, De Bellis G, Luiselli D, Brigidi P, Mabulla A, Marlowe F, Henry AG, Crittenden AN. 2014. Gut microbiome of the Hadza hunter-gatherers. Nat Commun 5:3654.

Tessler M, Neumann JS, Afshinnekoo E, Pineda M, Hersch R, Velho LFM, Segovia BT, Lansac-Toha FA, Lemke M, DeSalle R, Mason CE, Brugler MR. 2017. Large-scale differences in microbial biodiversity discovery between 16S amplicon and shotgun sequencing. Sci Rep 7:6589.

Tirosh A, Calay ES, Tuncman G, Claiborn KC, Inouye KE, Eguchi K, Alcala M, Rathaus M, Hollander KS, Ron I, Livne R, Heianza Y, Qi L, Shai I, Garg R, Hotamisligil GS. 2019. The short-chain fatty acid propionate increases glucagon and FABP4 production, impairing insulin action in mice and humans. Sci Transl Med 11. doi:10.1126/scitranslmed.aav0120

Truong DT, Franzosa EA, Tickle TL, Scholz M, Weingart G, Pasolli E, Tett A, Huttenhower C, Segata N. 2015. MetaPhlAn2 for enhanced metagenomic taxonomic profiling. Nat Methods 12:902–903.

Turnbaugh PJ, Hamady M, Yatsunenko T, Cantarel BL, Duncan A, Ley RE, Sogin ML, Jones WJ, Roe BA, Affourtit JP, Egholm M, Henrissat B, Heath AC, Knight R, Gordon JI. 2009a. A core gut microbiome in obese and lean twins. Nature 457:480–484.

Turnbaugh PJ, Ley RE, Mahowald MA, Magrini V, Mardis ER, Gordon JI. 2006. An obesity-associated gut microbiome with increased capacity for energy harvest. Nature 444:1027–1031.

Turnbaugh PJ, Ridaura VK, Faith JJ, Rey FE, Knight R, Gordon JI. 2009b. The effect of diet on the human gut microbiome: a metagenomic analysis in humanized gnotobiotic mice. Sci Transl Med 1:6ra14.

Vieira-Silva S, Falony G, Belda E, Nielsen T, Aron-Wisnewsky J, Chakaroun R, Forslund SK, Assmann K, Valles-Colomer M, Nguyen TTD, Proost S, Prifti E, Tremaroli V, Pons N, Le Chatelier E, Andreelli F, Bastard J-P, Coelho LP, Galleron N, Hansen TH, Hulot J-S, Lewinter C, Pedersen HK, Quinquis B, Rouault C, Roume H, Salem J-E, Søndertoft NB, Touch S, MetaCardis Consortium, Dumas M-E, Ehrlich SD, Galan P, Gøtze JP, Hansen T, Holst JJ, Køber L, Letunic I, Nielsen J, Oppert J-M, Stumvoll M, Vestergaard H, Zucker J-D, Bork P, Pedersen O, Bäckhed F, Clément K, Raes J. 2020. Statin therapy is associated with lower prevalence of gut microbiota dysbiosis. Nature 581:310–315.

Walter J, Armet AM, Finlay BB, Shanahan F. 2020. Establishing or Exaggerating Causality for the Gut Microbiome: Lessons from Human Microbiota-Associated Rodents. Cell 180:221–232.

WHO Expert Consultation. 2004. Appropriate body-mass index for Asian populations and its implications for policy and intervention strategies. Lancet 363:157–163.

Wu H, Esteve E, Tremaroli V, Khan MT, Caesar R, Mannerås-Holm L, Ståhlman M, Olsson LM, Serino M, Planas-Fèlix M, Xifra G, Mercader JM, Torrents D, Burcelin R, Ricart W, Perkins R, Fernàndez-Real JM, Bäckhed F. 2017. Metformin alters the gut microbiome of individuals with treatment-naive type 2 diabetes, contributing to the therapeutic effects of the drug. Nat Med 23:850–858.

Zemb O, Achard CS, Hamelin J, De Almeida M-L, Gabinaud B, Cauquil L, Verschuren LMG, Godon J-J. 2020. Absolute quantitation of microbes using 16S rRNA gene metabarcoding: A rapid normalization of relative abundances by quantitative PCR targeting a 16S rRNA gene spike-in standard. Microbiologyopen 9:e977.

Zhang X, Zhong H, Li Y, Shi Z, Ren H, Zhang Z, Zhou X, Tang S, Han X, Lin Y, Yang F, Wang D, Fang C, Fu Z, Wang L, Zhu S, Hou Y, Xu X, Yang H, Wang J, Kristiansen K, Li J, Ji L. 2021. Sex- and age-related trajectories of the adult human gut microbiota shared across populations of different ethnicities. Nature Aging 1:87–100.

Zheng W, McLerran DF, Rolland B, Zhang X, Inoue M, Matsuo K, He J, Gupta PC, Ramadas K, Tsugane S, Irie F, Tamakoshi A, Gao Y-T, Wang R, Shu X-O, Tsuji I, Kuriyama S, Tanaka H, Satoh H, Chen C-J, Yuan J-M, Yoo K-Y, Ahsan H, Pan W-H, Gu D, Pednekar MS, Sauvaget C, Sasazuki S, Sairenchi T, Yang G, Xiang Y-B, Nagai M, Suzuki T, Nishino Y, You S-L, Koh W-P, Park SK, Chen Y, Shen C-Y, Thornquist M, Feng Z, Kang D, Boffetta P, Potter JD. 2011. Association between body-mass index and risk of death in more than 1 million Asians. N Engl J Med 364:719–729.

Zouiouich S, Loftfield E, Huybrechts I, Viallon V, Louca P, Vogtmann E, Wells PM, Steves CJ, Herzig K-H, Menni C, Jarvelin M-R, Sinha R, Gunter MJ. 2021. Markers of metabolic health and gut microbiome diversity: findings from two population-based cohort studies. Diabetologia 64:1749–1759.